

# Beyond the walled garden.

*A visual essay in five chapters.*

*Jo Wood*

# Preface

*The Walled Garden*

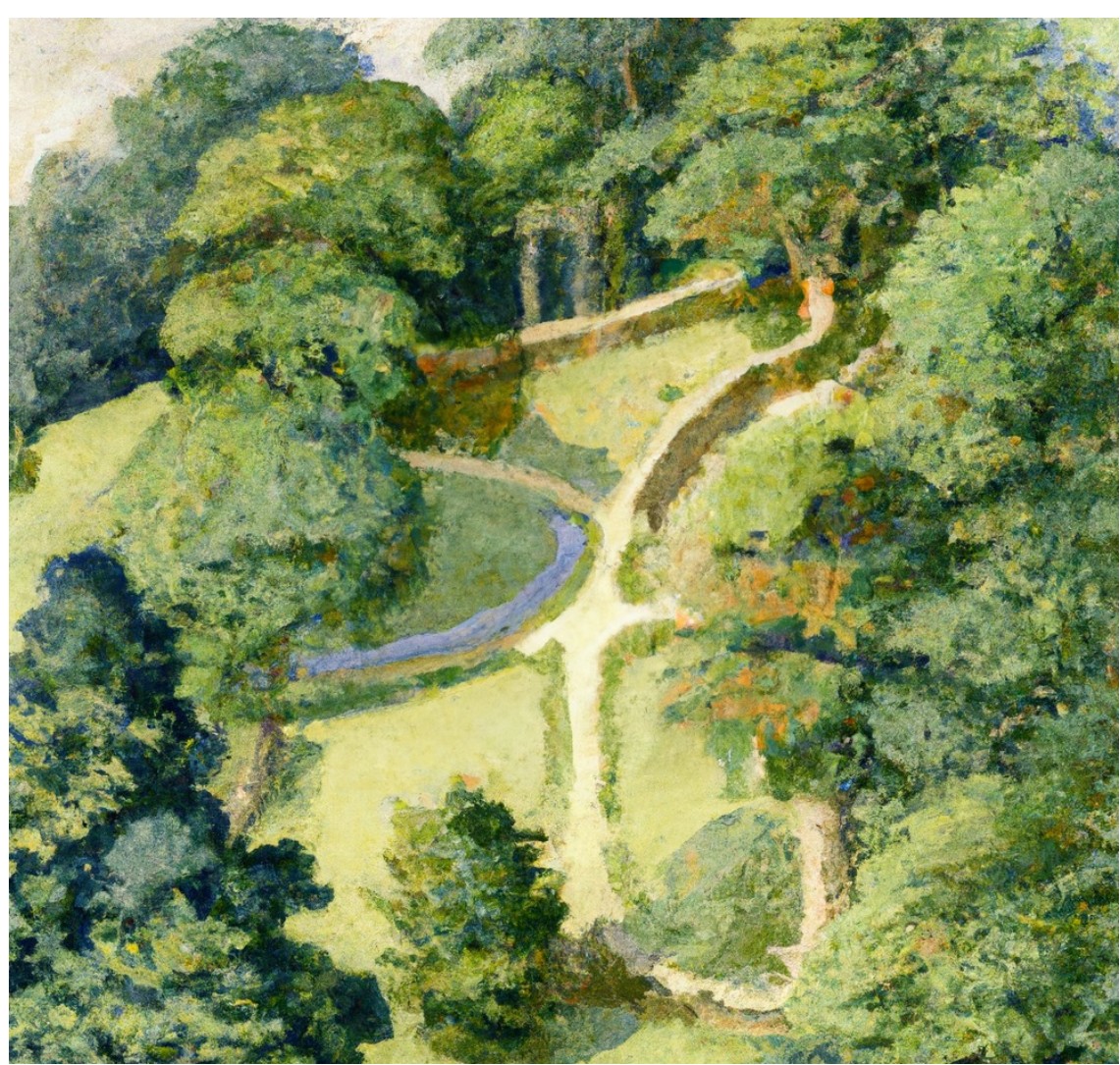

Data visualization was tended in a walled garden.

It was a familiar and comforting place. Grass surrounded by a border planted with flowers, a shady corner, a formal ornamental patch. A short path led to a nursery where new plants were grown before being planted in the garden.

The garden was surrounded by trees. They too were comforting. Those who worked in the garden felt protected. They knew their place. They *belonged*.

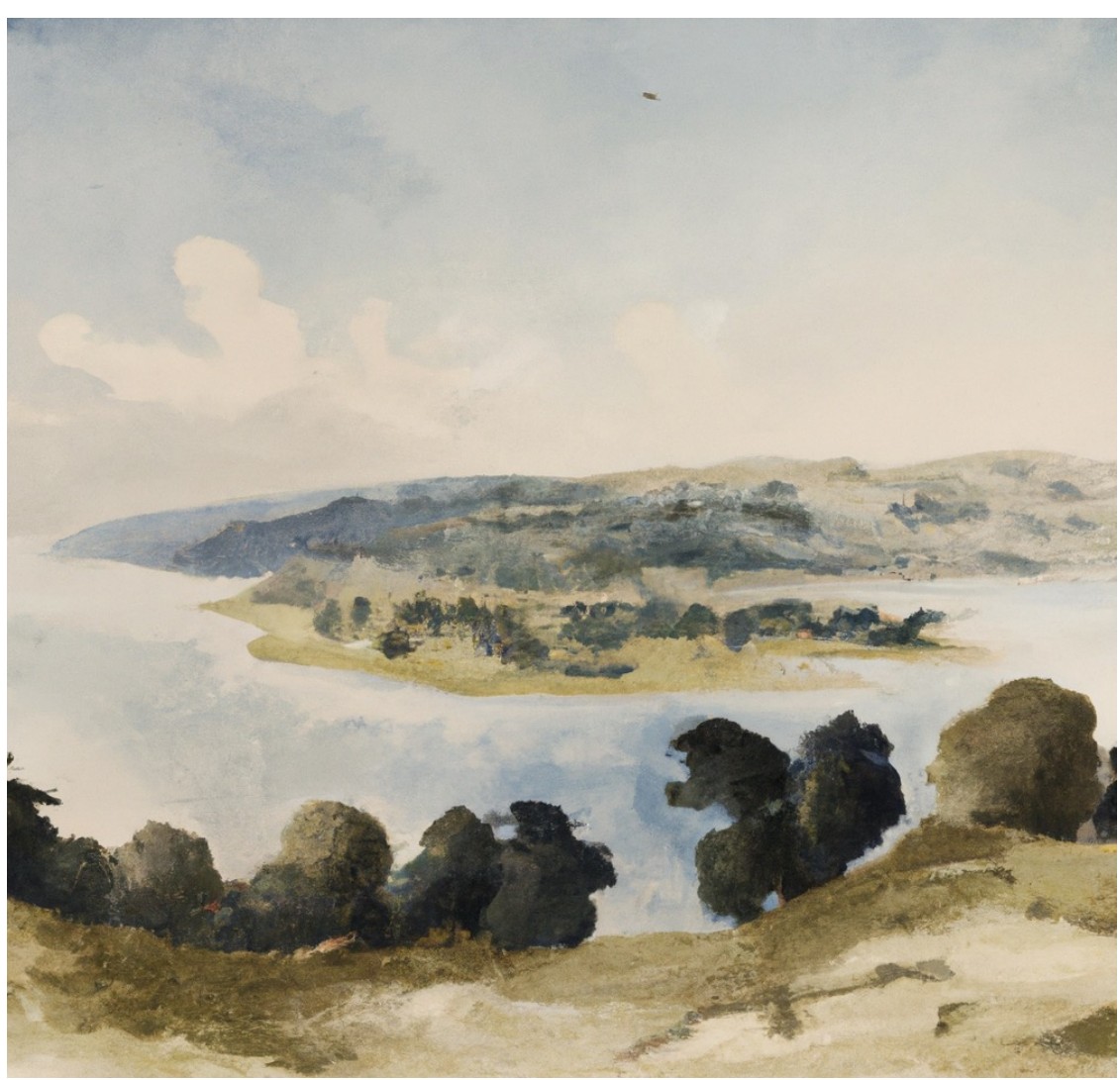

Yet there was land beyond the trees.

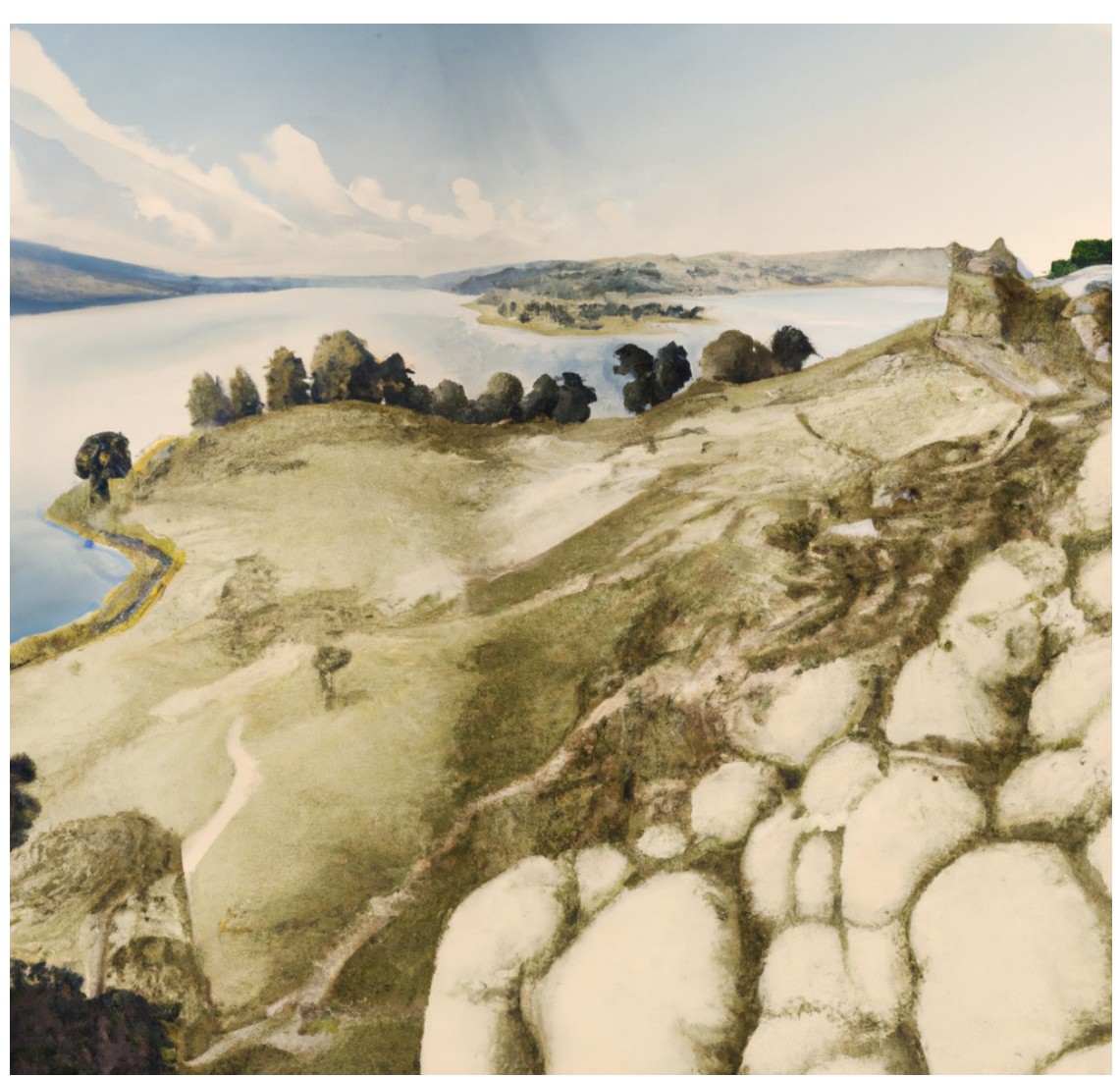

A land both familiar and unfamiliar.

It wasn't cultivated like the garden. Plants grew wild. Hills were adorned with cascades of rocks.  Lakes of unknown depth and creatures unimagined. The land was unruly and beautiful.

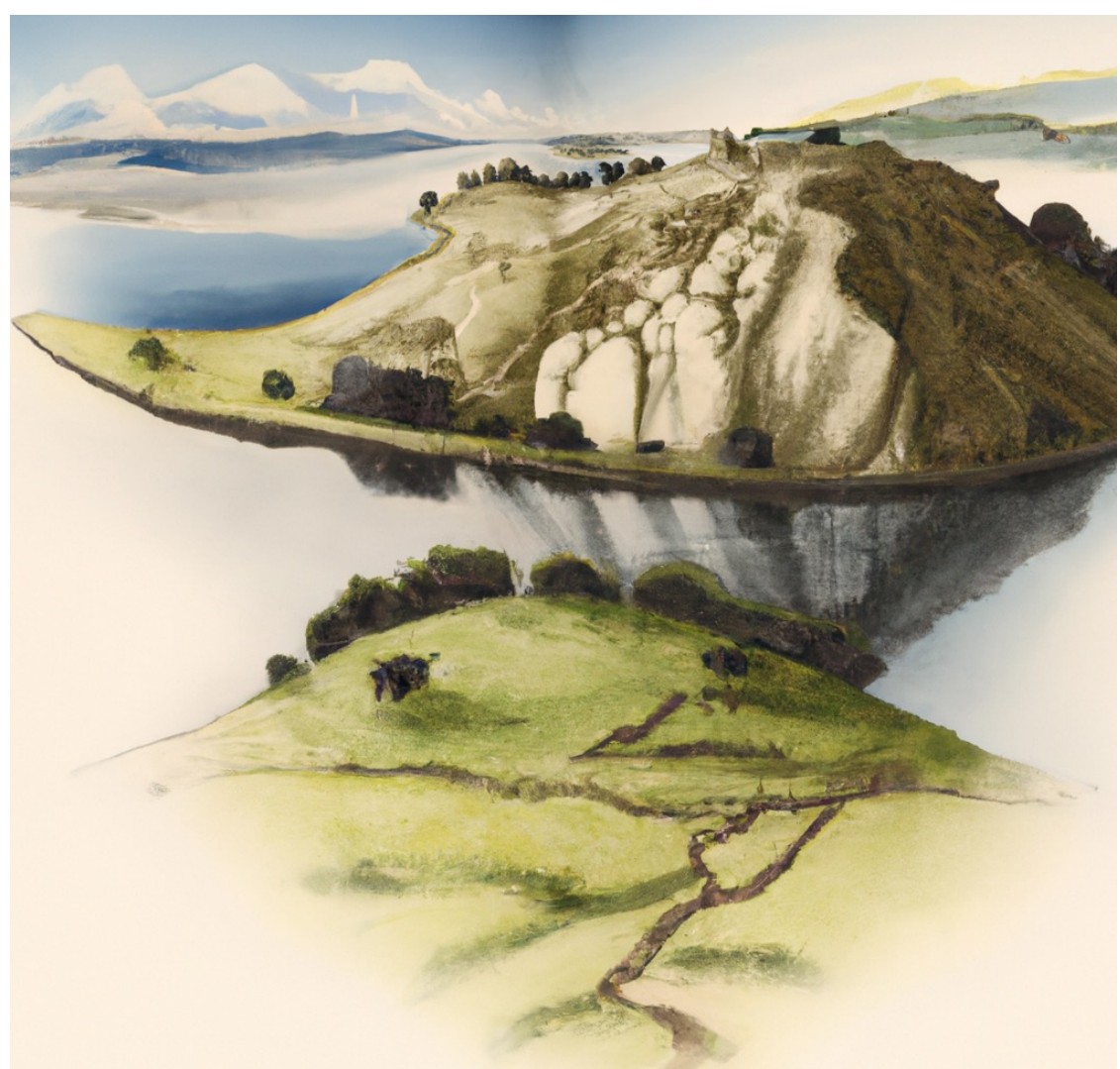

Beyond the water were new places. Mountains yet to be explored.

Landscapes without walls.

# Chapter I

*Atomisation of design*

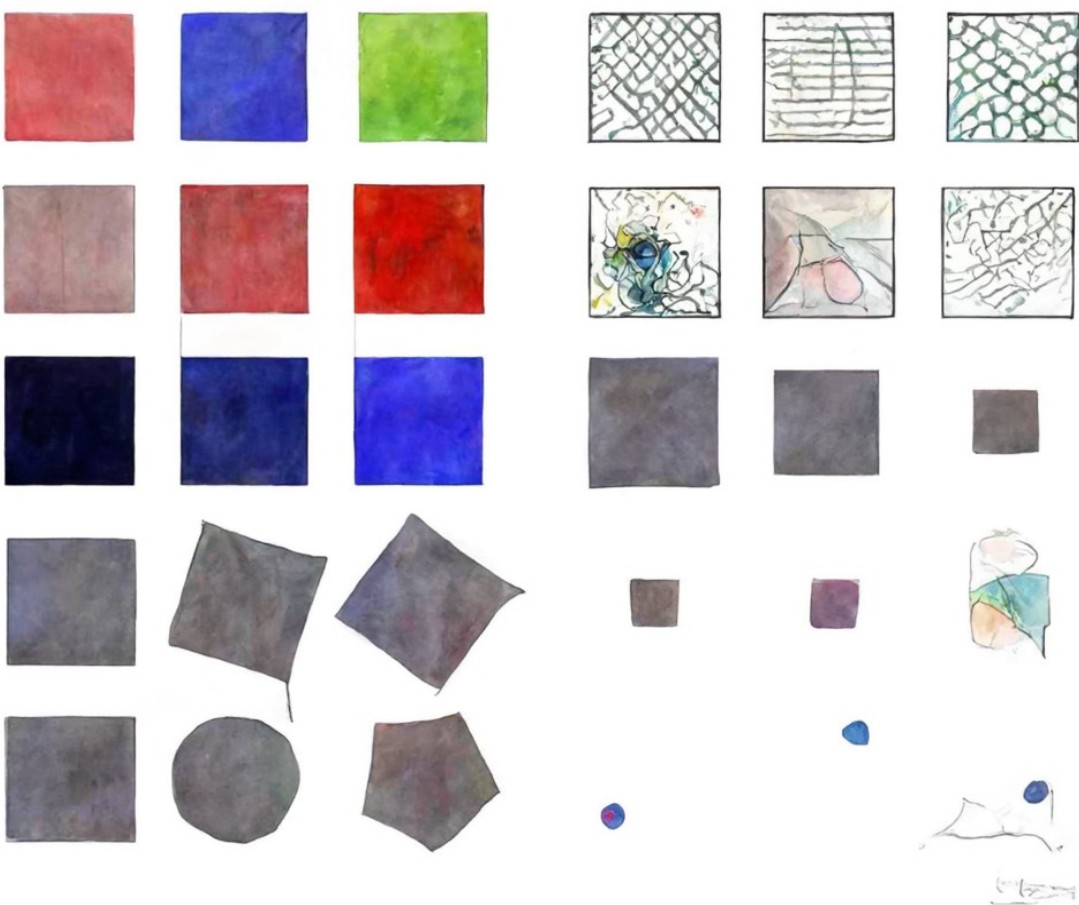

It is in the nature of scientific endeavour to atomise our understanding. We strive to control, to isolate and to abstract. Bertin (1983) sought to apply this approach to design. He argued that if we can disaggregate design into atomic retinal variables and we are able to associate distinct objects with those variables, we can reassemble them in whatever combination suits our task.

Visual design becomes an activity of *decomposition*, *combination* and *reaggregation*.

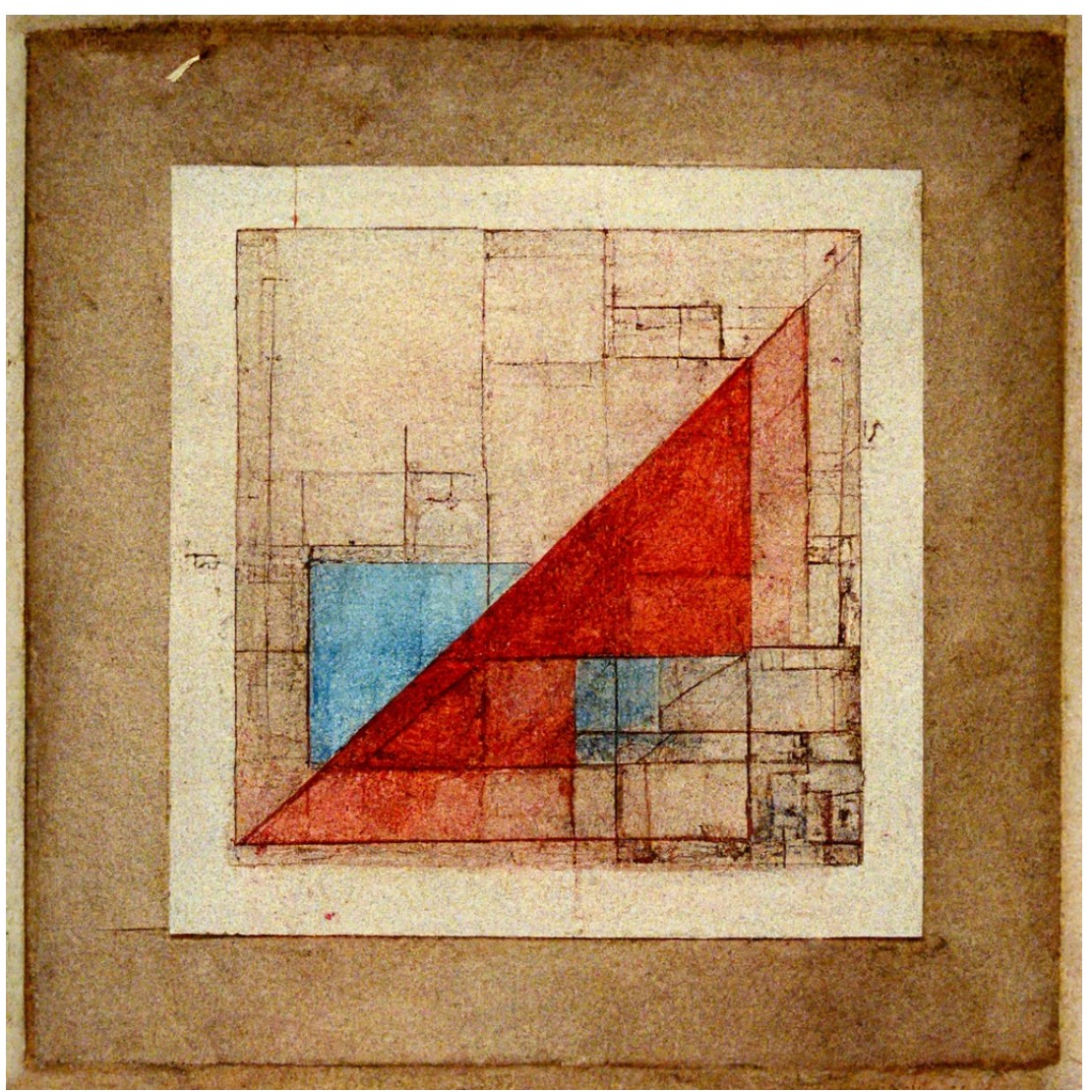

The appeal of this approach is understandable. We abstract and simplify the world. We develop precise and controlled visual languages with which to express our abstractions.

And we hope designer and reader alike share a common understanding of that graphical language.

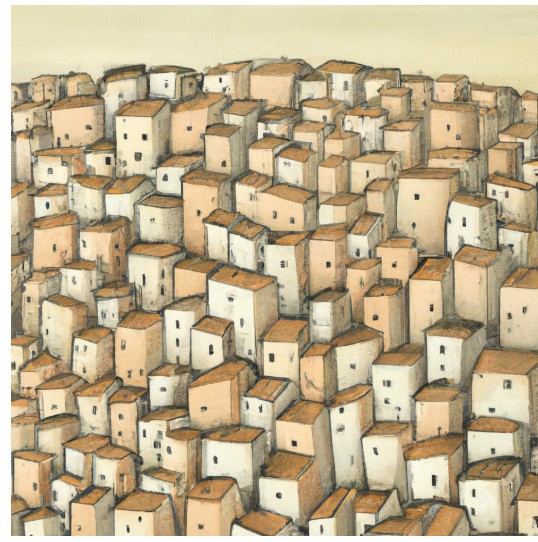
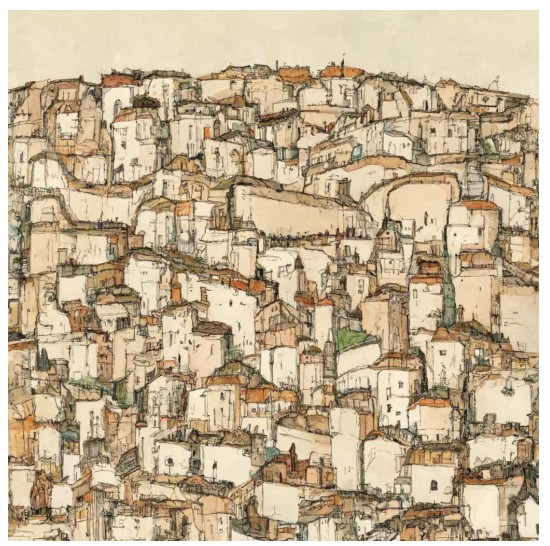
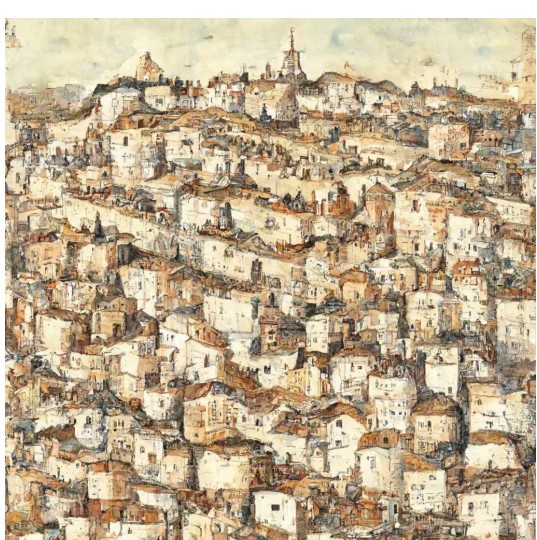
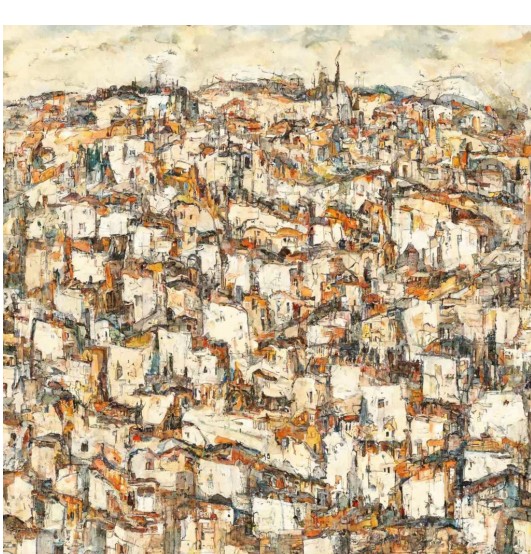

But it also confines us to a walled garden of definable objects and abstract geometric shapes sequentially modified in distinct data-driven channels.

How can we express more subtle and intertwined character in our visual languages?

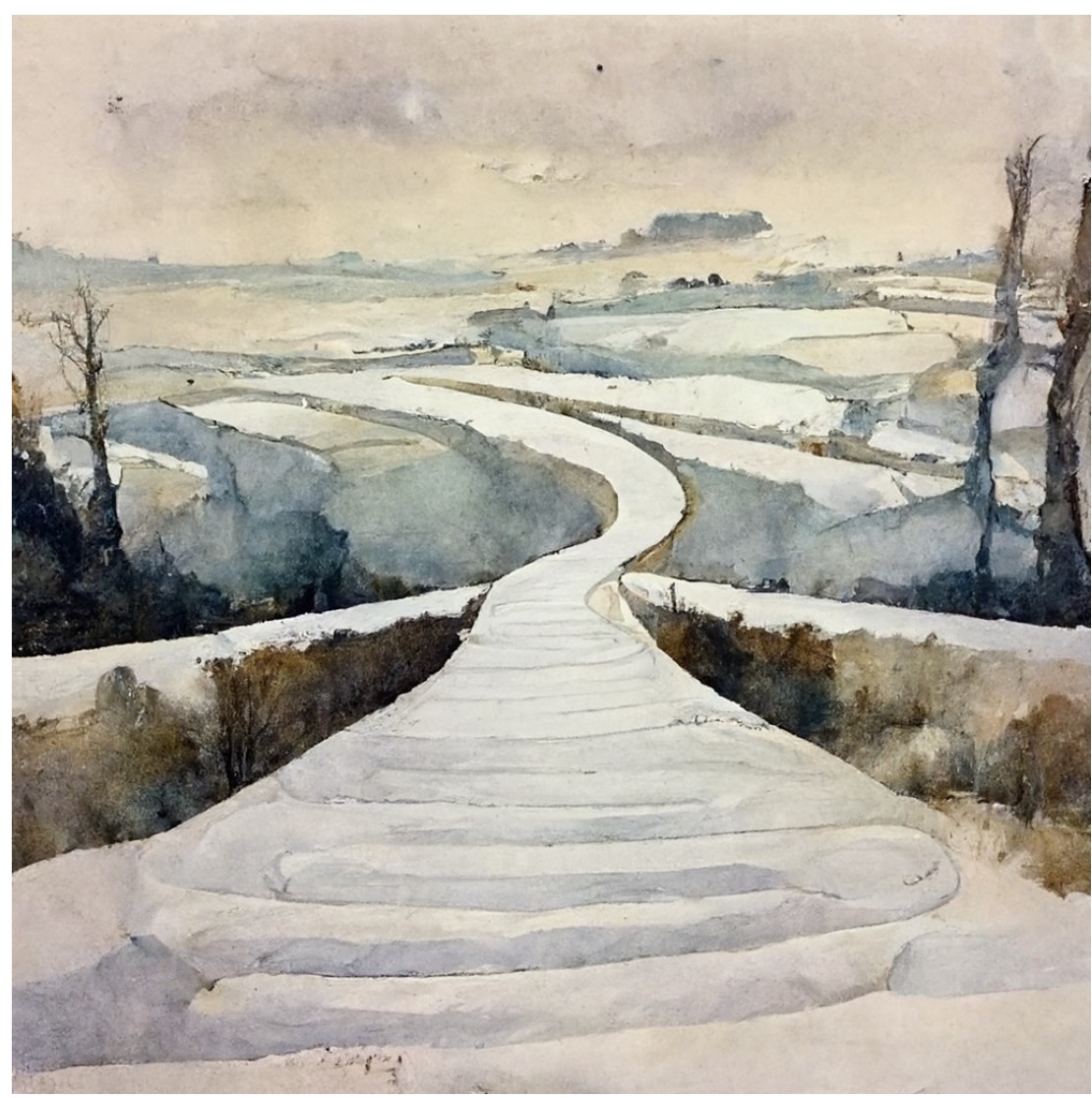

Instead of 'hue' or 'orientation' or 'size' why shouldn't data visualization designers be able to express data by 'sadness' or 'vertigo' or 'naivety'? Instead of depicting 'locatable objects' (Mackinlay, 1986), why shouldn't we shape our designs more holistically?

In short, we need more expressive pathways along which we explore visual design space.

# Chapter II

*Expression and Effect*

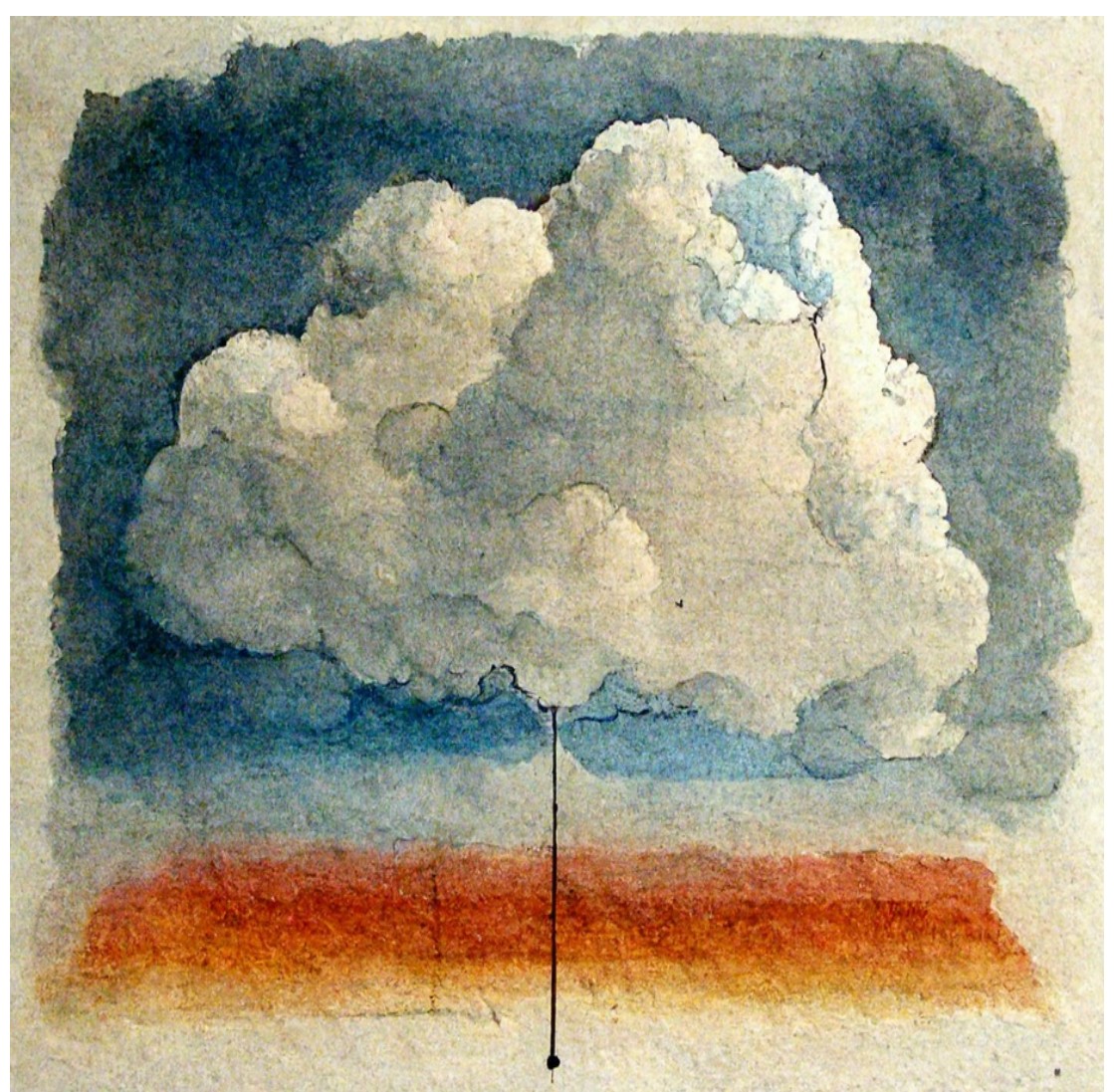

Jock Mackinlay (1986) characterised visualization as a process of visual encoding and decoding via the notions of *expressiveness* and *effectiveness*. A transformation of information into and out of the visual realm via a graphical language.

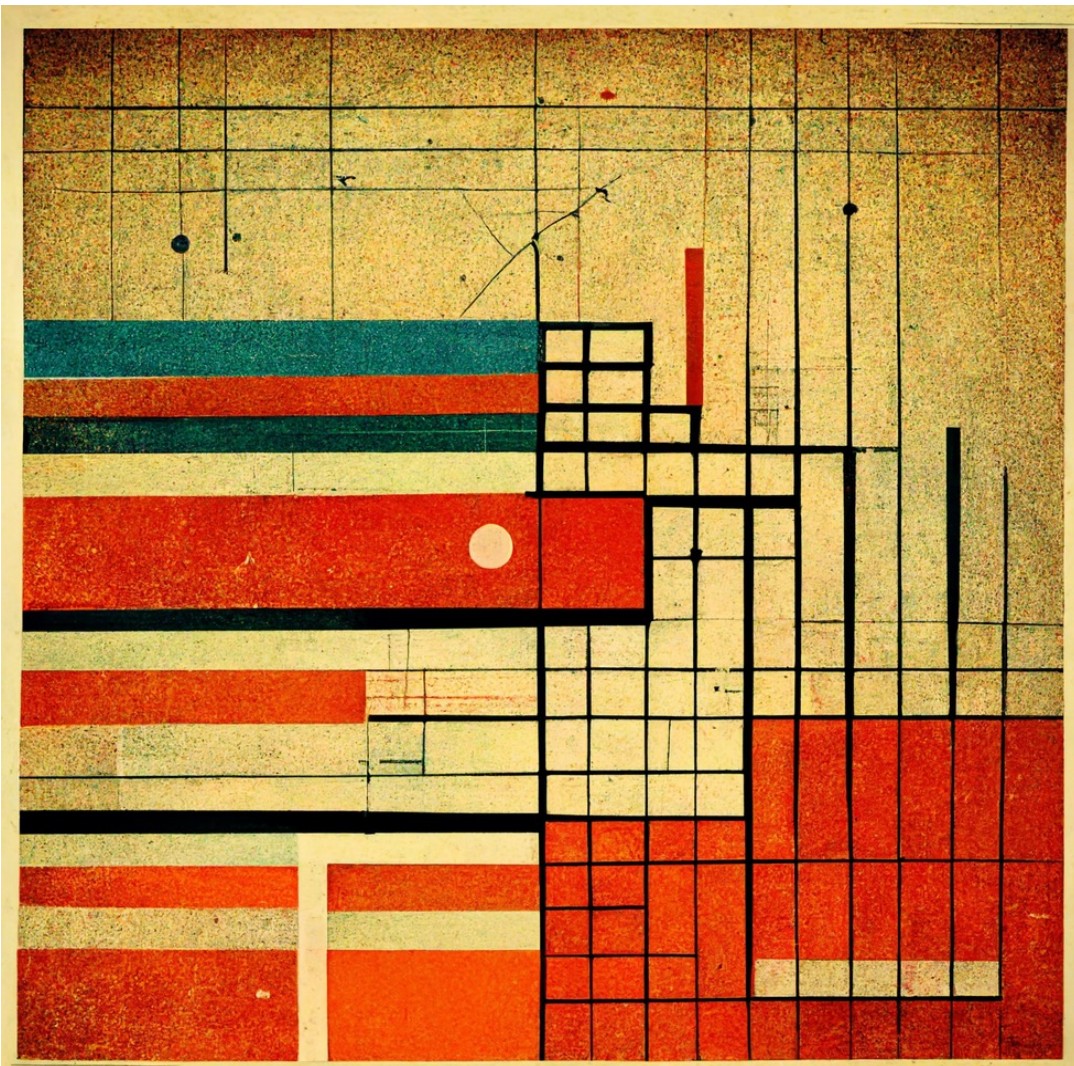

Visualization is *effective* when we are able to decode a graphic language and infer the communicative intent of the designer. This is the principle that governed much of the foundational empirical work of Cleveland and McGill (1984).

That process is most reliable when that language is well understood by all parties. Consequently, in data visualization we tend toward simple, less ambiguous graphical convention.

Geometric primitives, Cartesian spaces, fixed colour palettes.

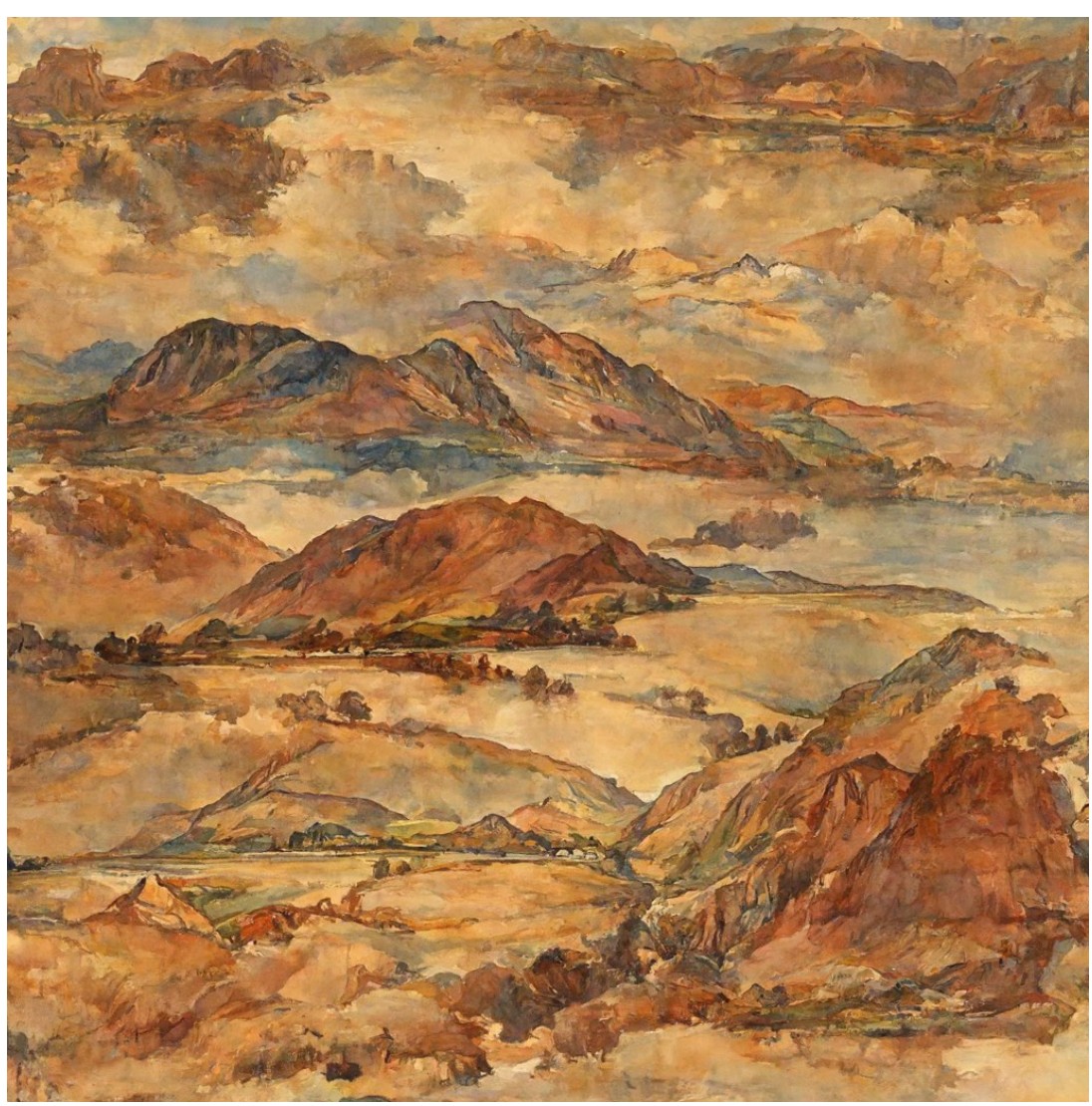

Yet in the visual arts we are more *expressive* using a much richer more diverse visual vocabulary to encode artistic intent.

Emotion, nuance, introspection, ambiguity, contradiction.

That expressiveness arrives bearing a cost. With a more sophisticated language, we demand more of the reader in their ability to decode the subtleties of visual languages.

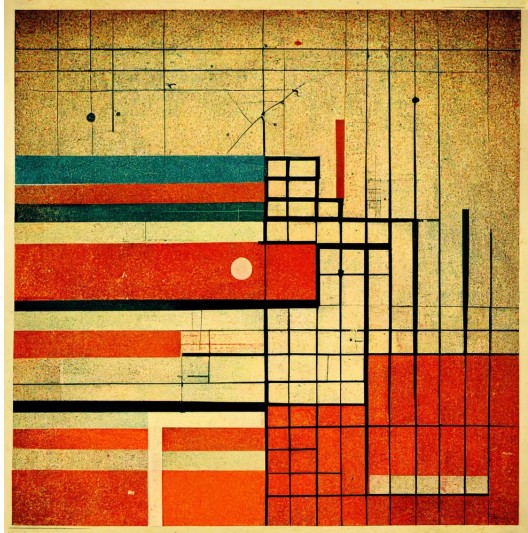
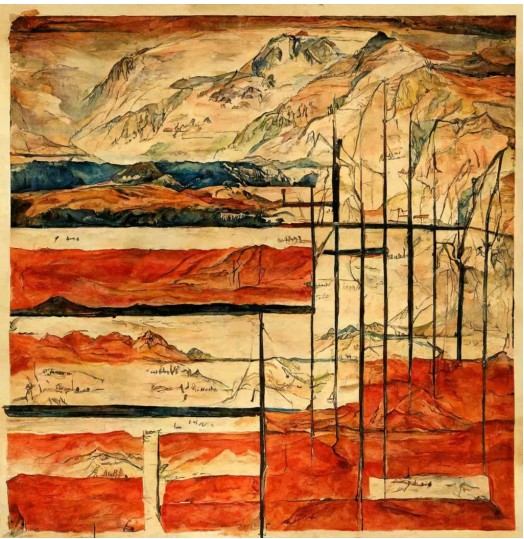
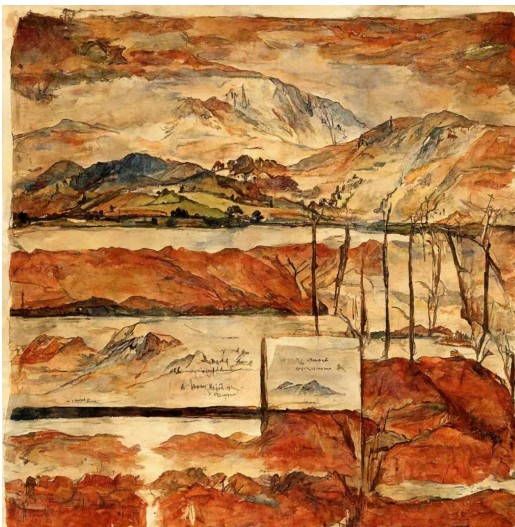
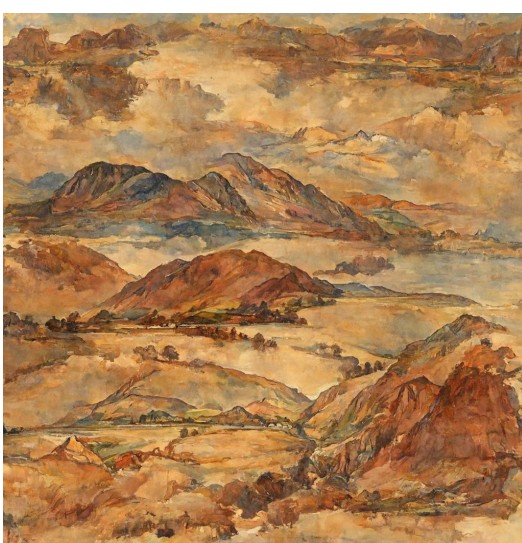

The creator of a visual artifact experiences a tension between simplicity to support effectiveness and sophistication to support expressiveness.

That design tension places their work on a continuum.

A continuum that new technologies may be opening up to many more of us in new and surprising ways.

# Chapter III

*A new technology*

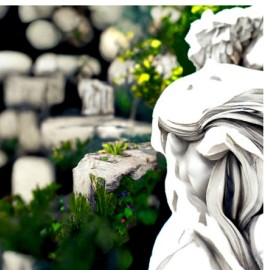

*"A marble Greek statue in the middle of a Doric garden"*

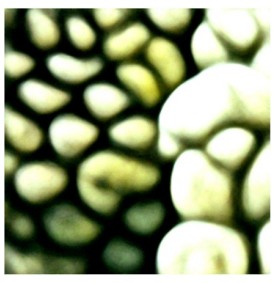

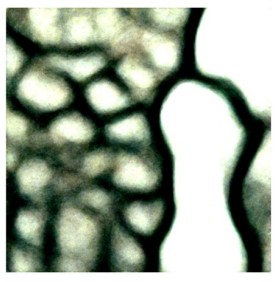

The last two years have seen step change in the sophistication with which textual language that describes images may be modelled. Contrastive Language-Image Pretraining (*CLIP*) provides new ways to access vast repositories of graphical artifacts with natural language.

Machine effectiveness.

Large image collections can be aggregated and recombined via a *diffusion* process in a combinatorial explosion of possibility.

Machine expressiveness.

Together the CLIP-diffusion process (Kim et al 2021) encourages us to guide image creation with both natural and graphical languages.

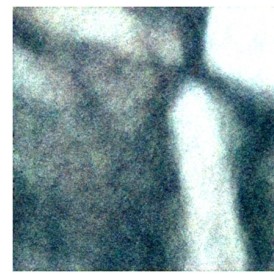

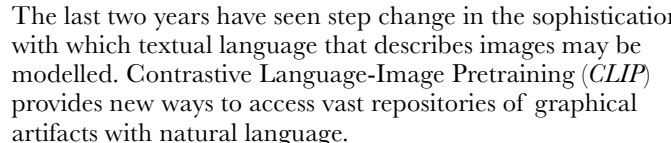

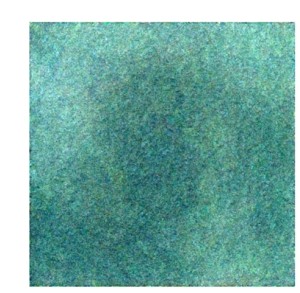

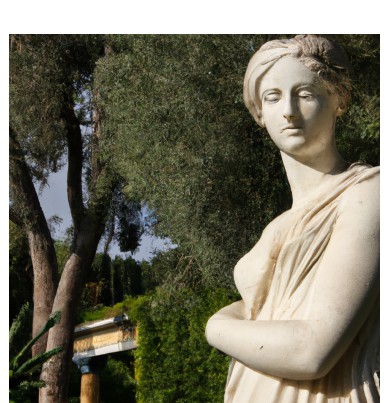

*"…in spring with fields of daffodils."*

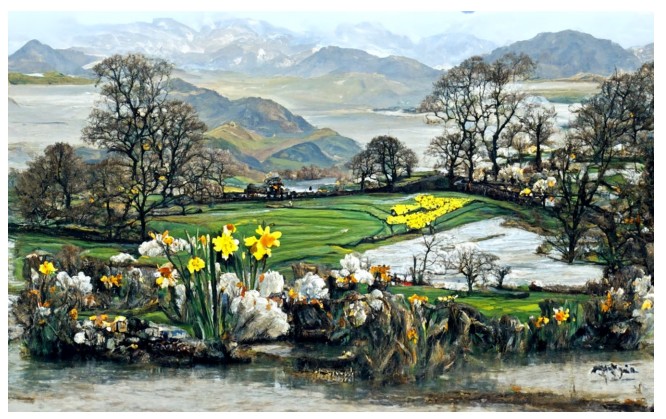

*"…on a bright summer day."*

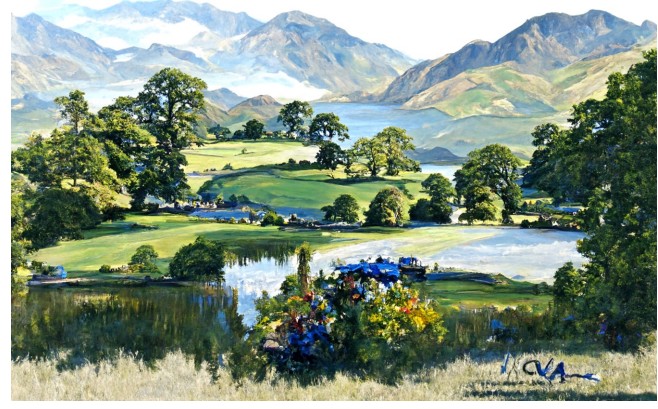

*"An oil painting of a scenic view of the English Lake District…"*

*"…in rich autumn colours."*

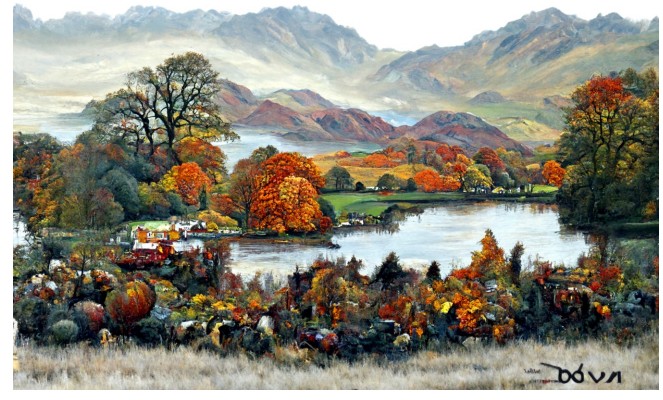

*"…in the depths of a snowy winter."*

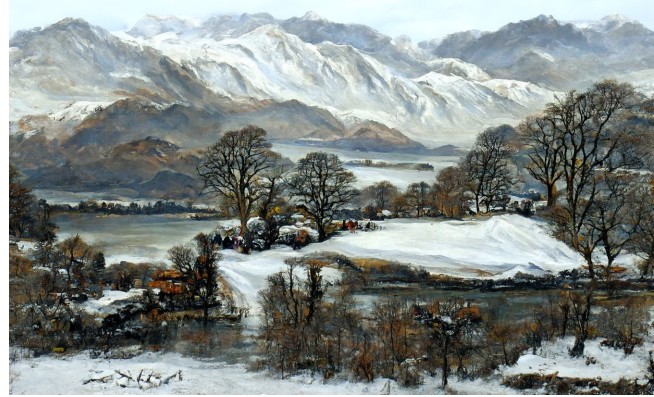

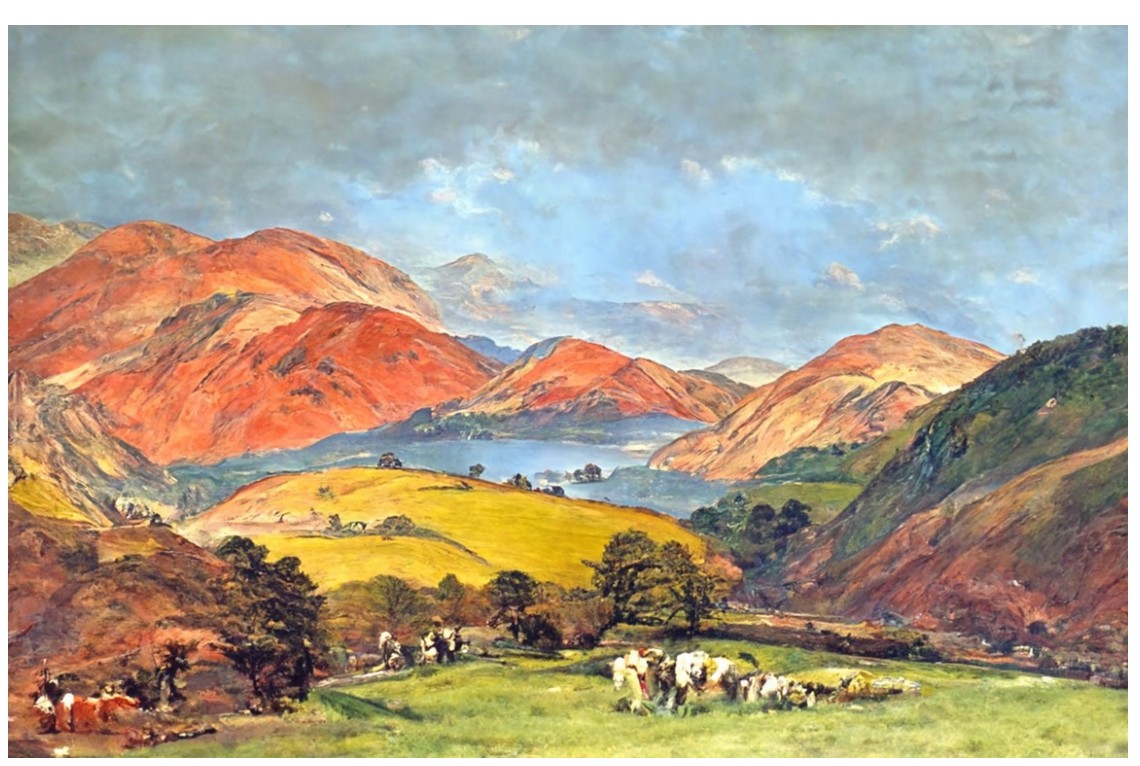

We may exercise expressiveness not only with natural language but by seeding image generation with our own graphics.

This may be the image as a whole…

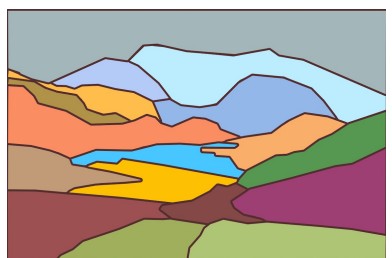

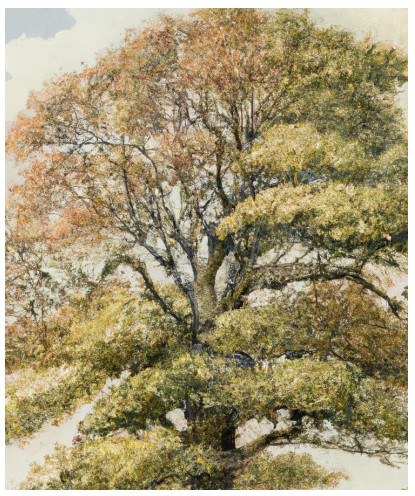

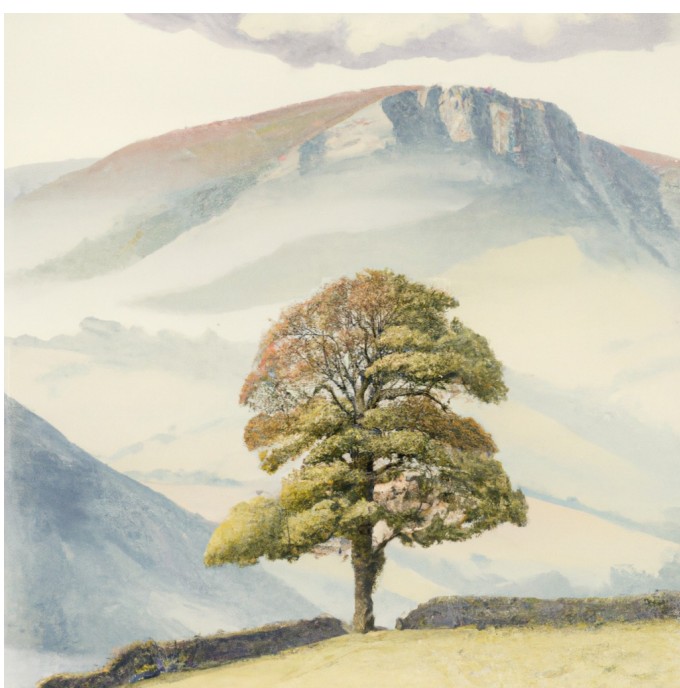

…or just part of it (*inpainting* and *outpainting*).

# Chapter IV

*But is it data visualization?*

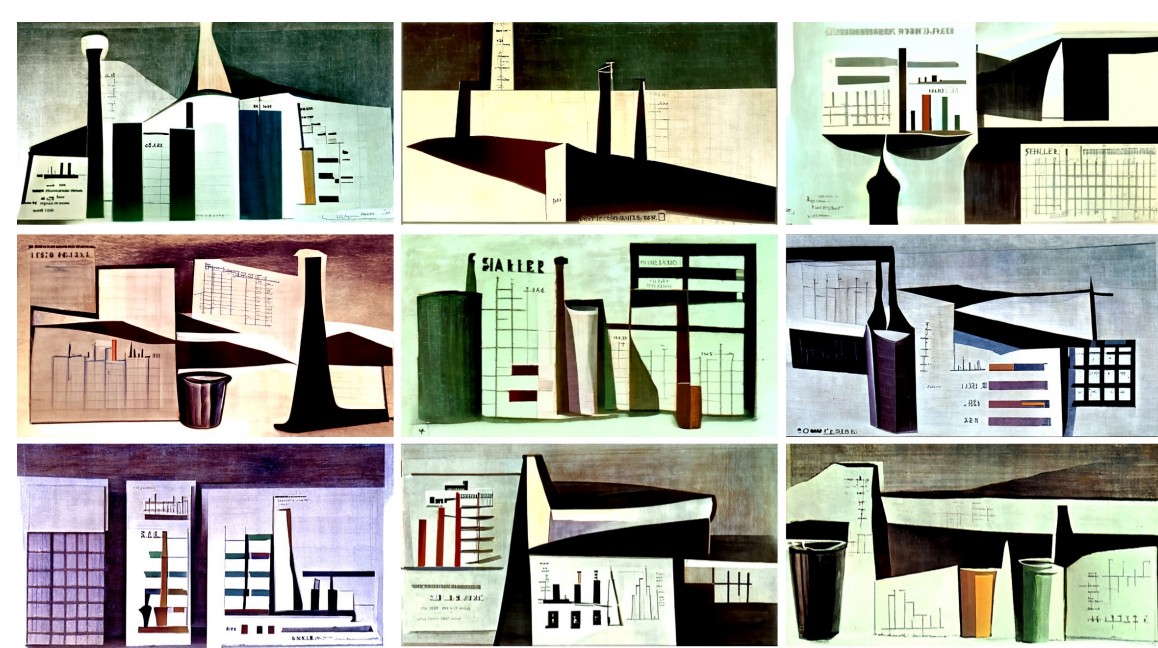

We might argue that these new text-to-image technologies are simply tools for pastiche (*"a lonely hotel in the style of Edward Hopper"*), or fleetingly amusing incongruities (the *"avocado armchair"*), or cliché (*"busy spaceport on an alien planet"*).

They may instead present us with a future of more expressive data visualization. Where we are no longer constrained by "locatable objects" and atomic "retinal variables".

But in gaining expressiveness, might we lose their effectiveness, rendering our graphics devoid of meaningful interpretation?

What follows are some examples that might suggest how we can move data visualization into new more expressive terrain.

They are data visualizations in that their character is still shaped by data. They don't rely on the designer possessing the skills of an artist to be expressive and could be generated at scale.

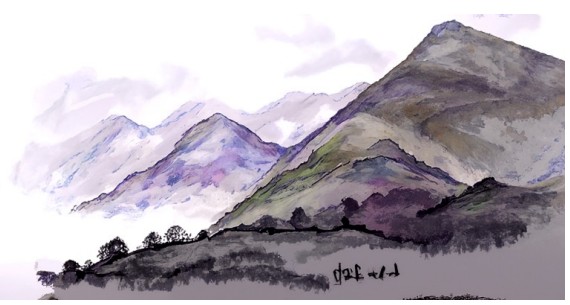
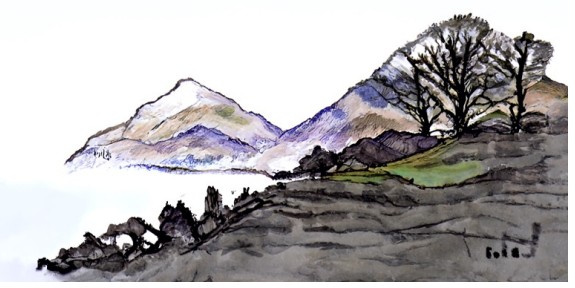
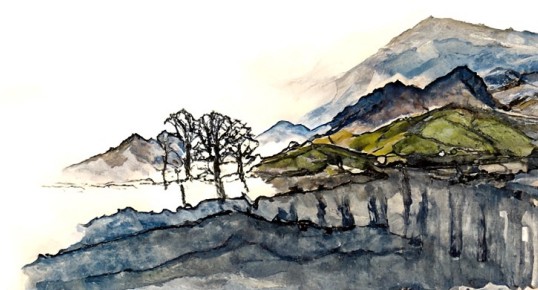
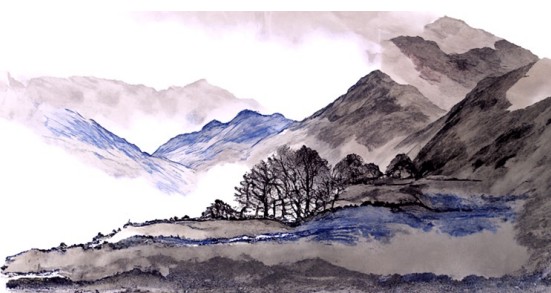

We might, for example, consider that expressive visualization allows us to generate emotional affect. Suppose we wished to show data *about* emotion. By weighting linguistic expression of sentiment in the CLIP process, we control how strongly it is expressed in visual form.

We start with four 'neutral' images from the same visual and textual seed: *"The hills and fells of the English Lake District drawn in watercolour and ink"*

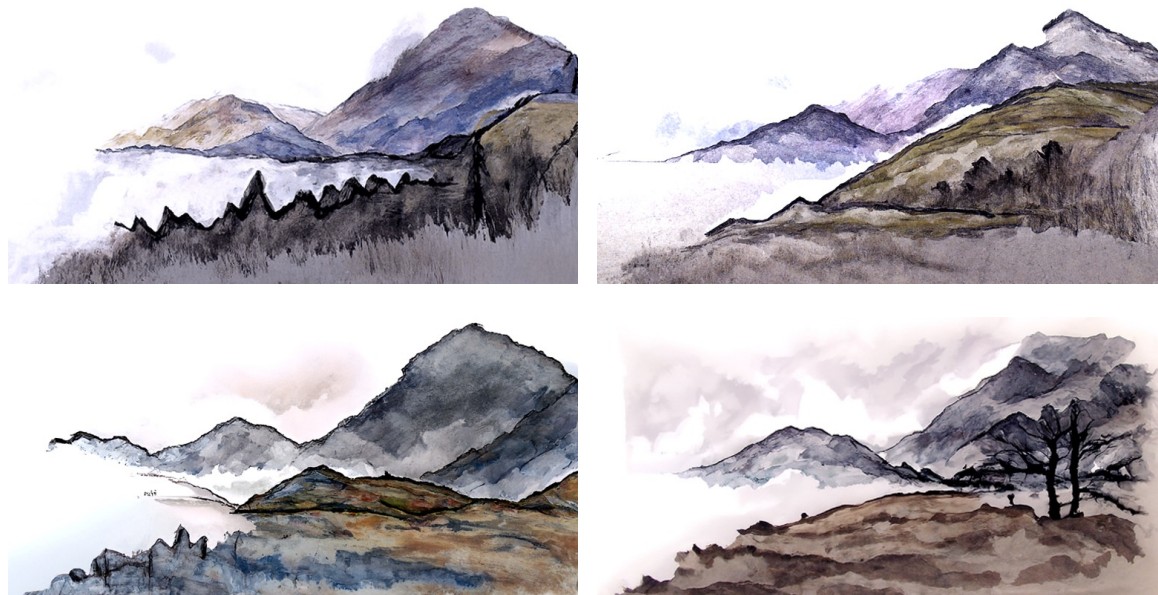

The phrase *"angry and fearful"* is mixed with the textual prompt that generates the image with a weighting of 25%.

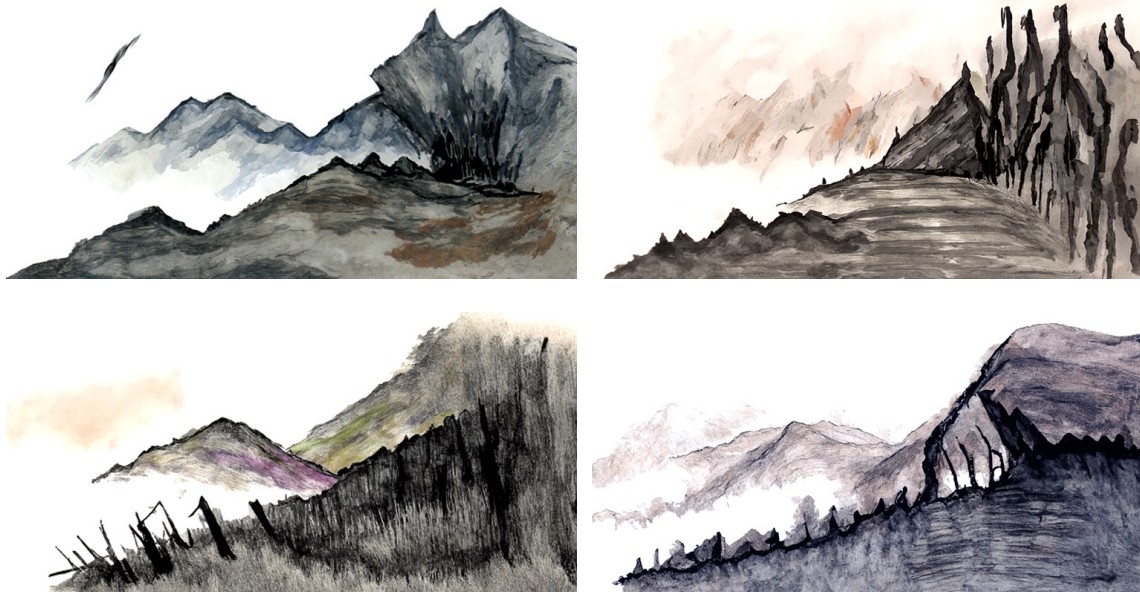

50% *"angry and fearful"*

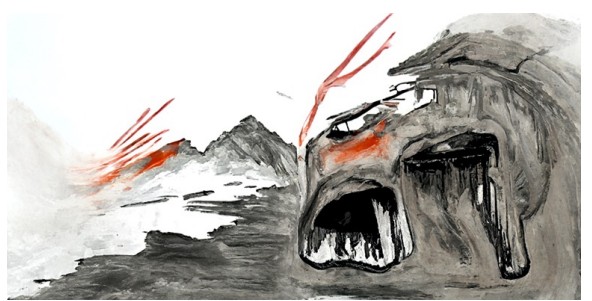
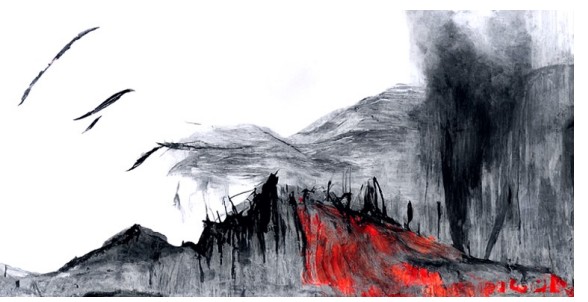
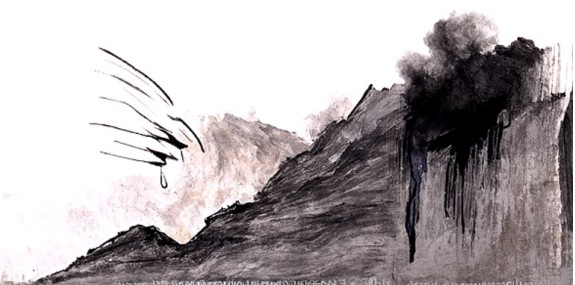
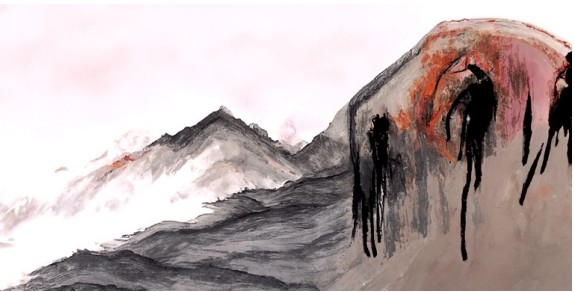

75% *"angry and fearful"*

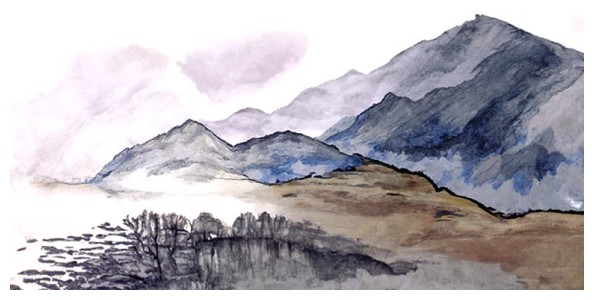
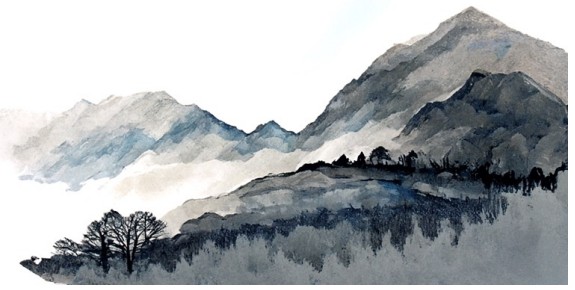
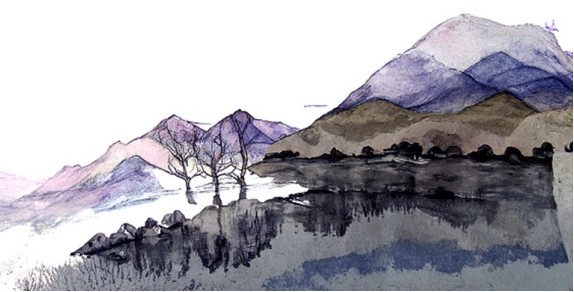
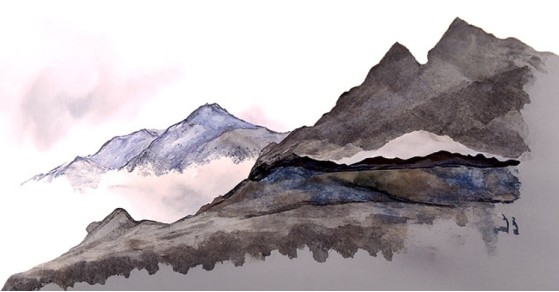

A similar exercise with a different sentiment expressed. Can you identify what it is?

25% sentiment weighting.

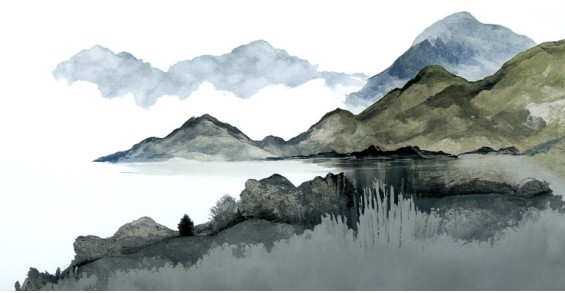
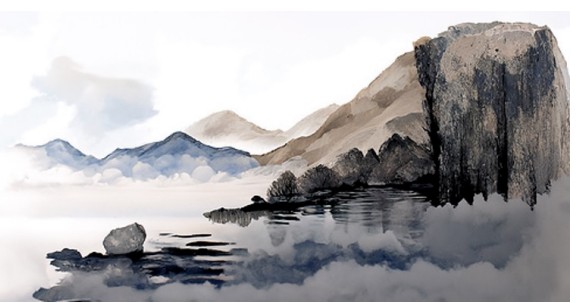
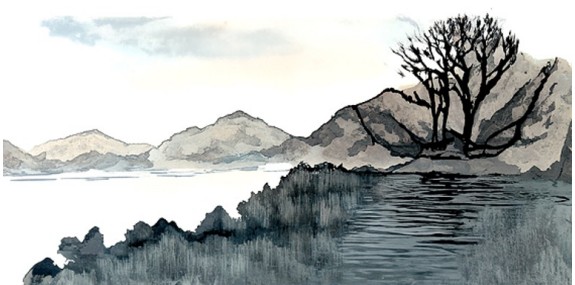
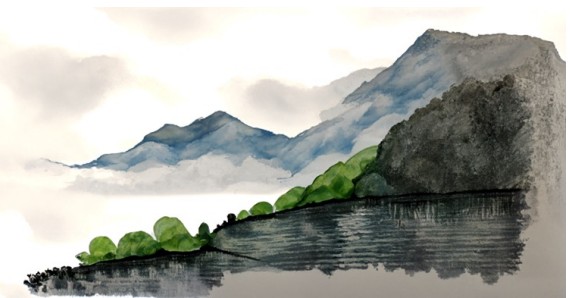

50% sentiment weighting.

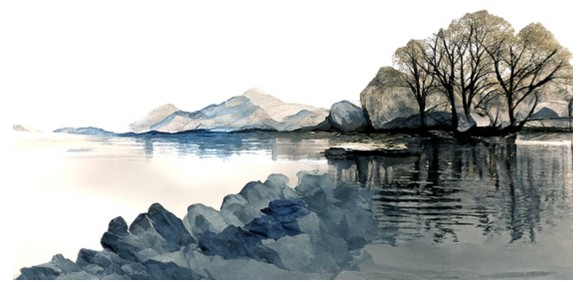

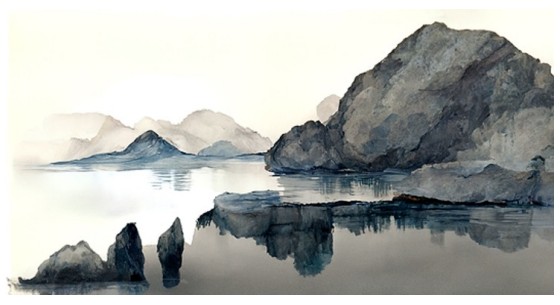

75% sentiment weighting.

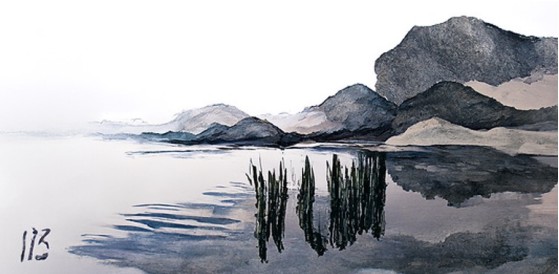

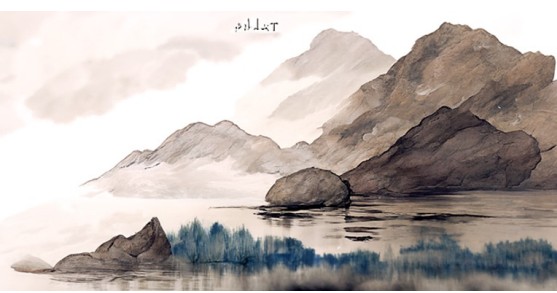

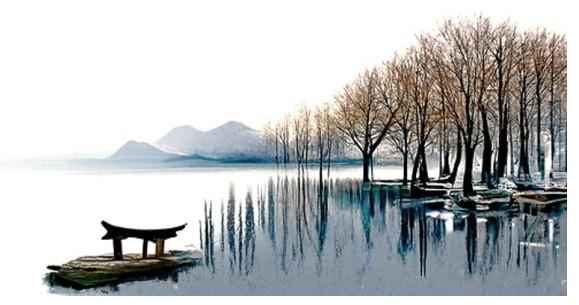
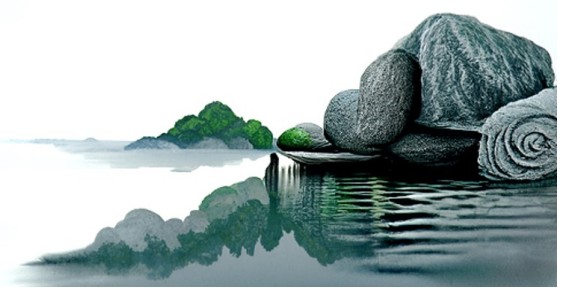

99% sentiment weighting.

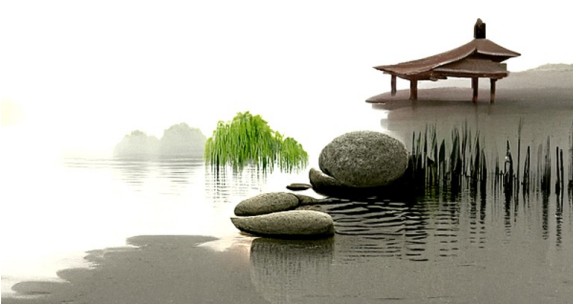
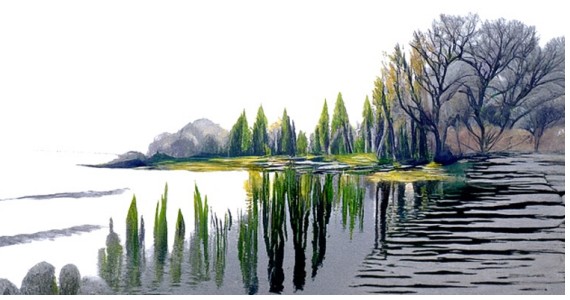

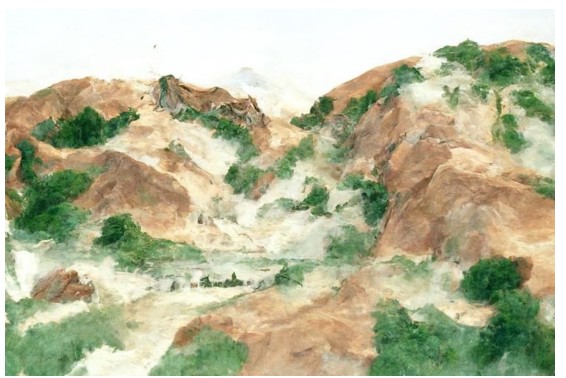
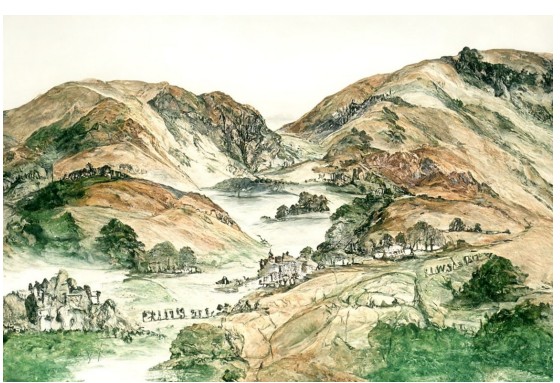
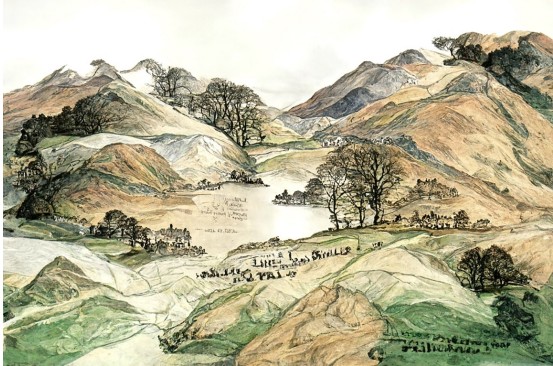
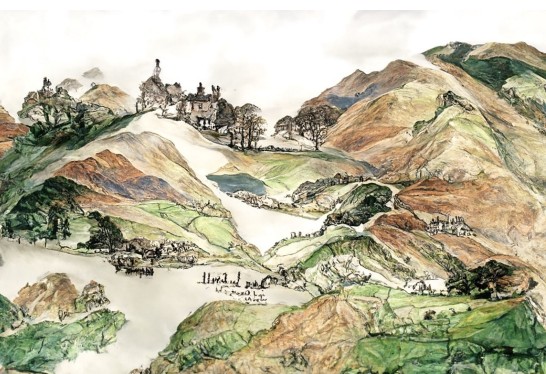

The relative strength of the image and text prompts that guide CLIP diffusion process and the 'speed' of movement through multi-dimension parameter space can itself be data-driven. This offers intriguing possibilities for new 'channels' in the mapping of data to graphic.

We might, for example, use guidance strength to represent uncertainty.

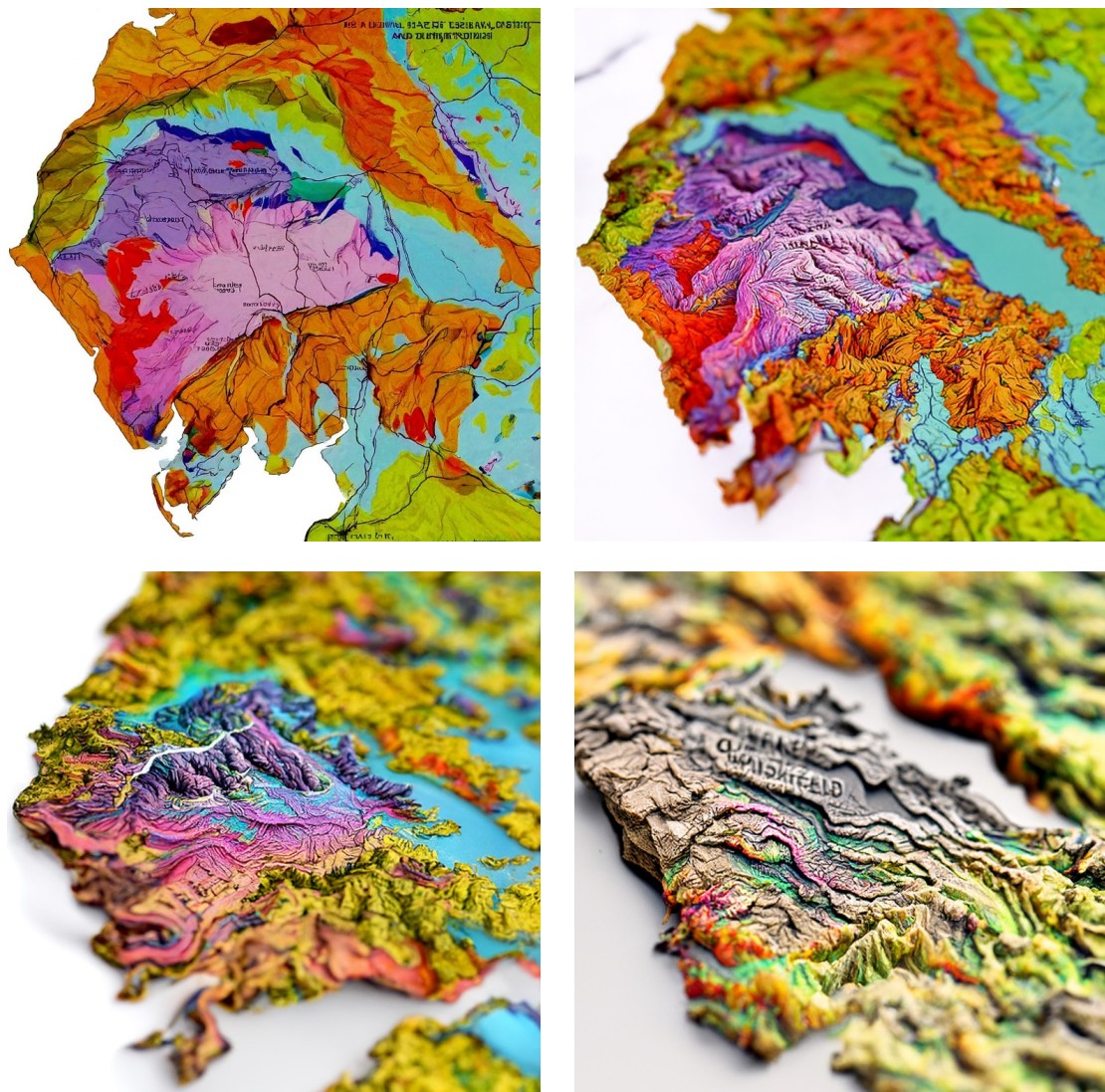

Greater expressiveness allows us to blend figurative and abstract symbolisation in new ways.

What if a geological map were to look like the rocks it symbolised?

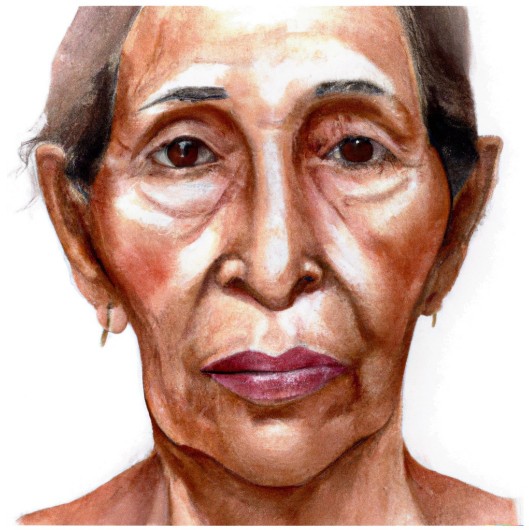
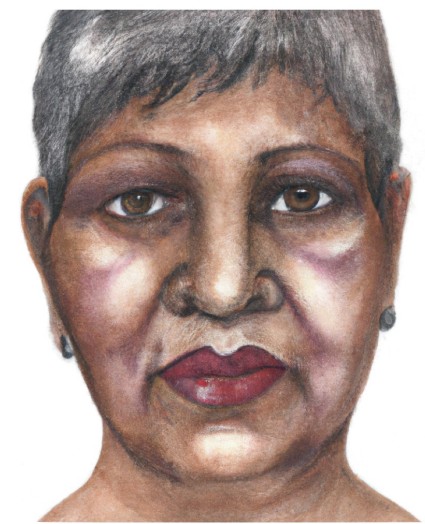
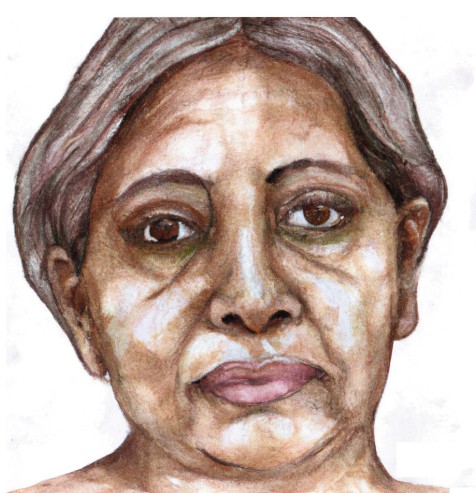
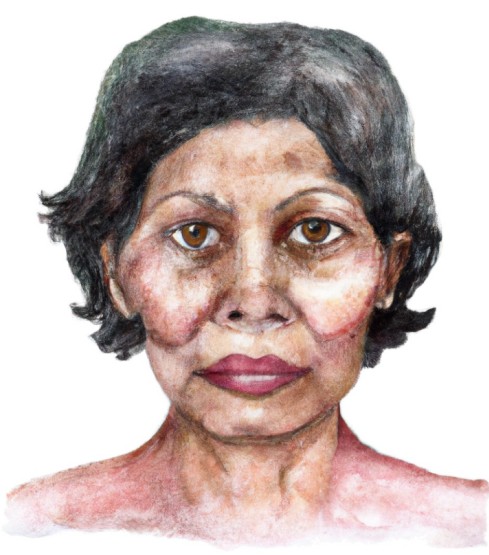

When *Chernoff faces* were proposed in 1973 for showing multivariate data, it was thought our ability to pre-attentively assess facial features might allow us to quickly process abstract face symbols carrying multivariate data (face shape, smiles/frowns, eye size etc.). In practice the limited graphical expressiveness of the time meant they met with little success.

Perhaps their time has now come.

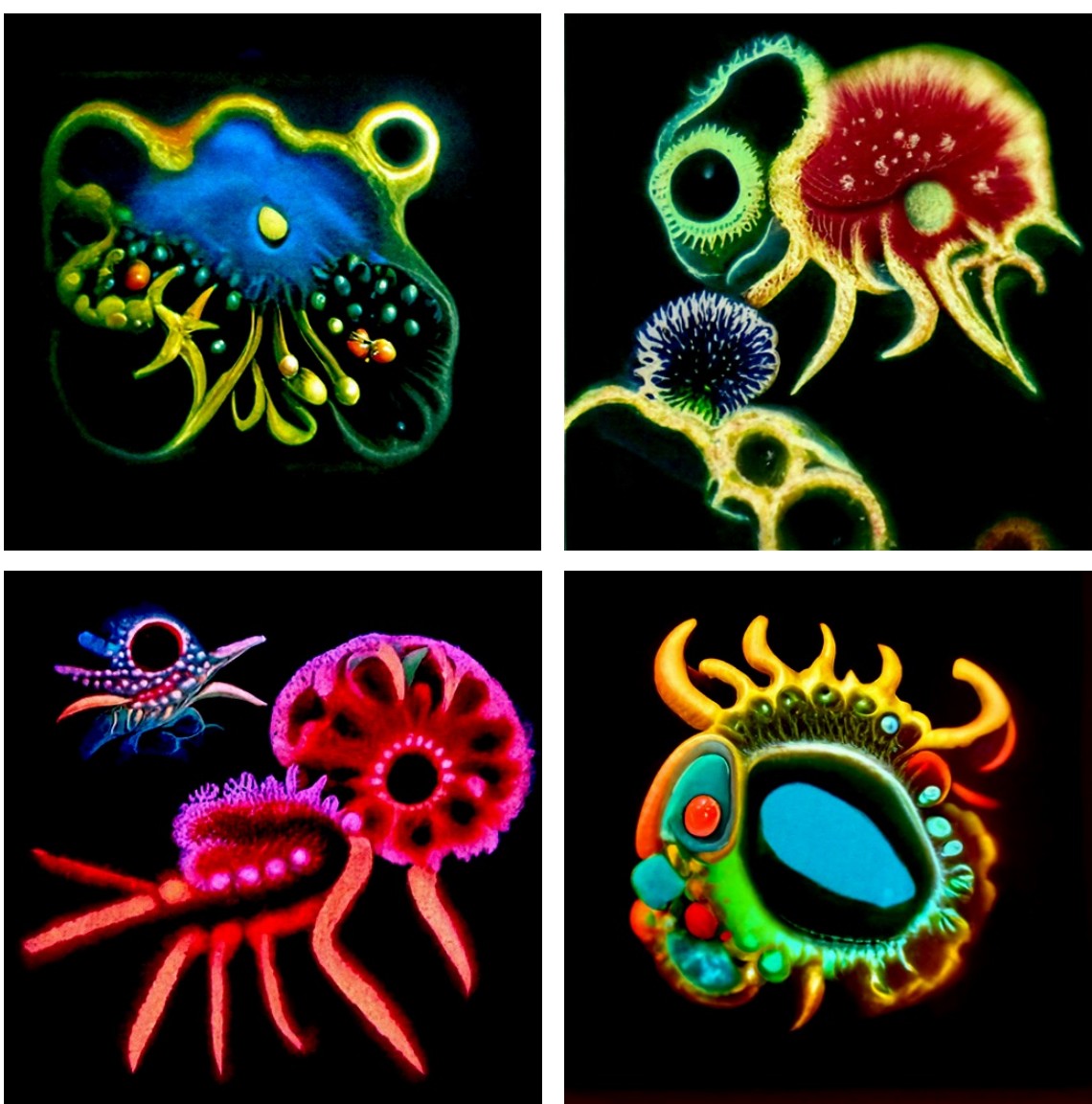

Or perhaps expressive glyph visualization moves us into new uncharted waters in which swim more memorable data creatures.

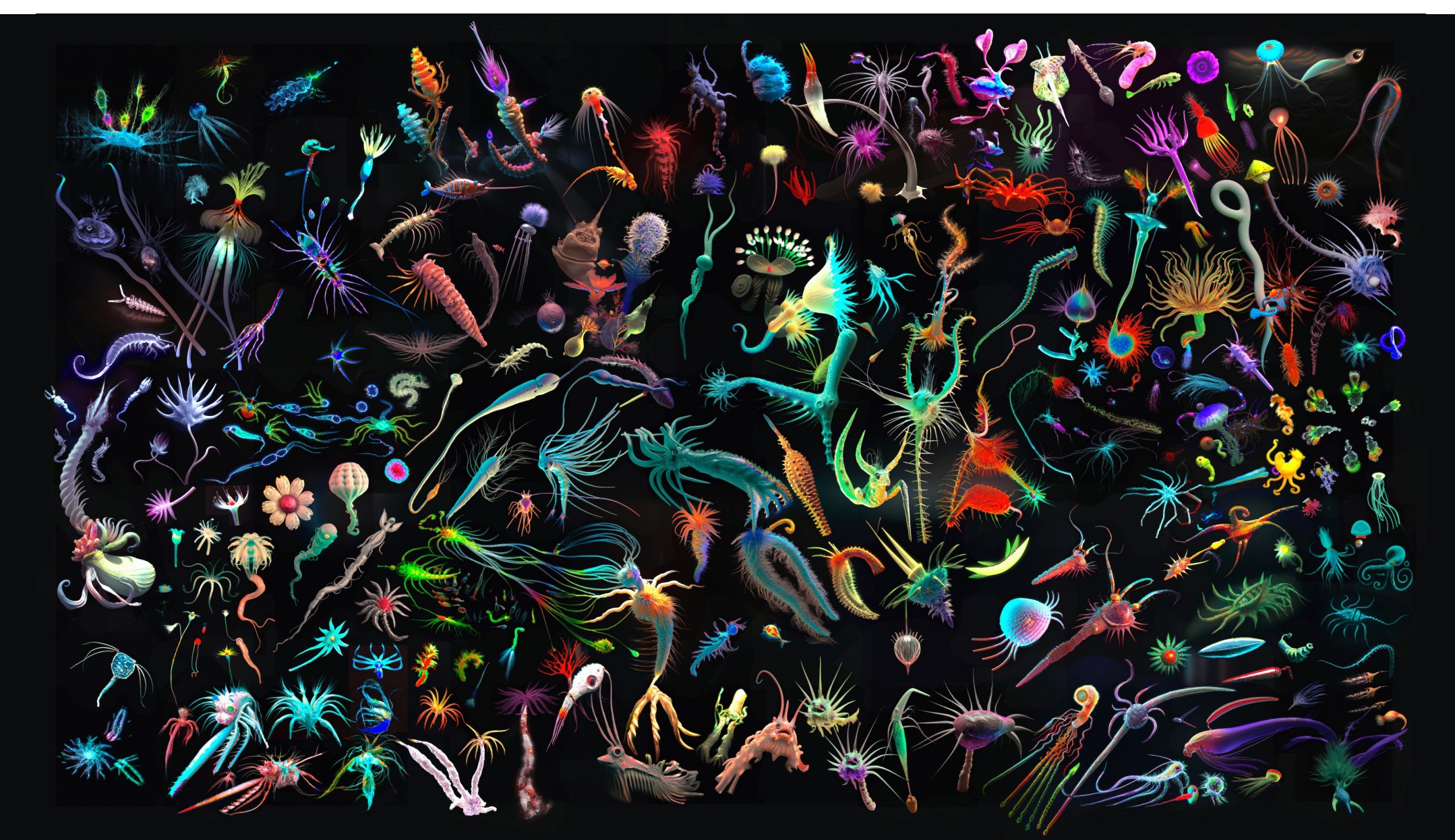

# Chapter V

*New landscapes*

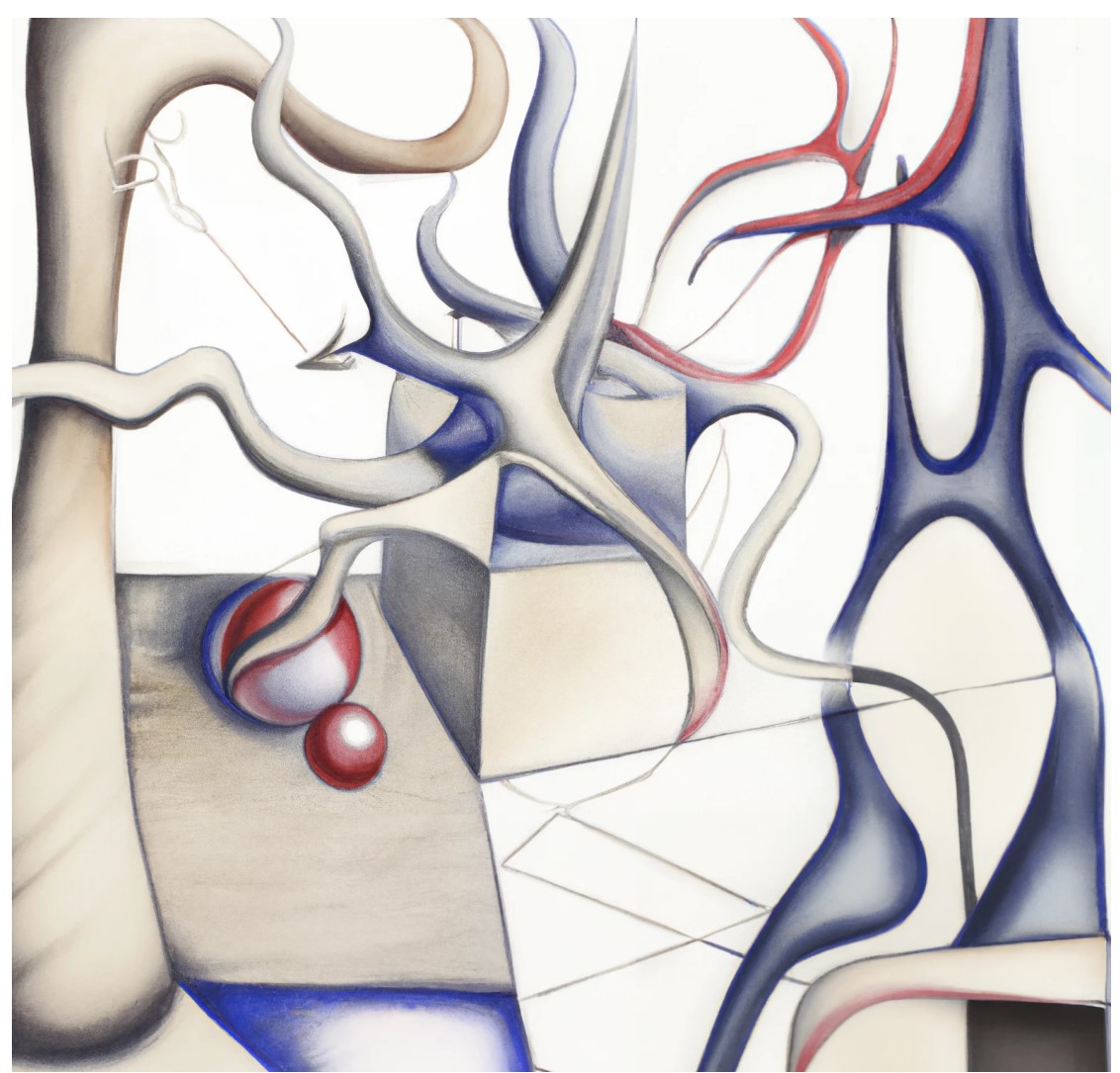

*"Visual ambiguity occupying the liminal space between abstract and figurative representation."*

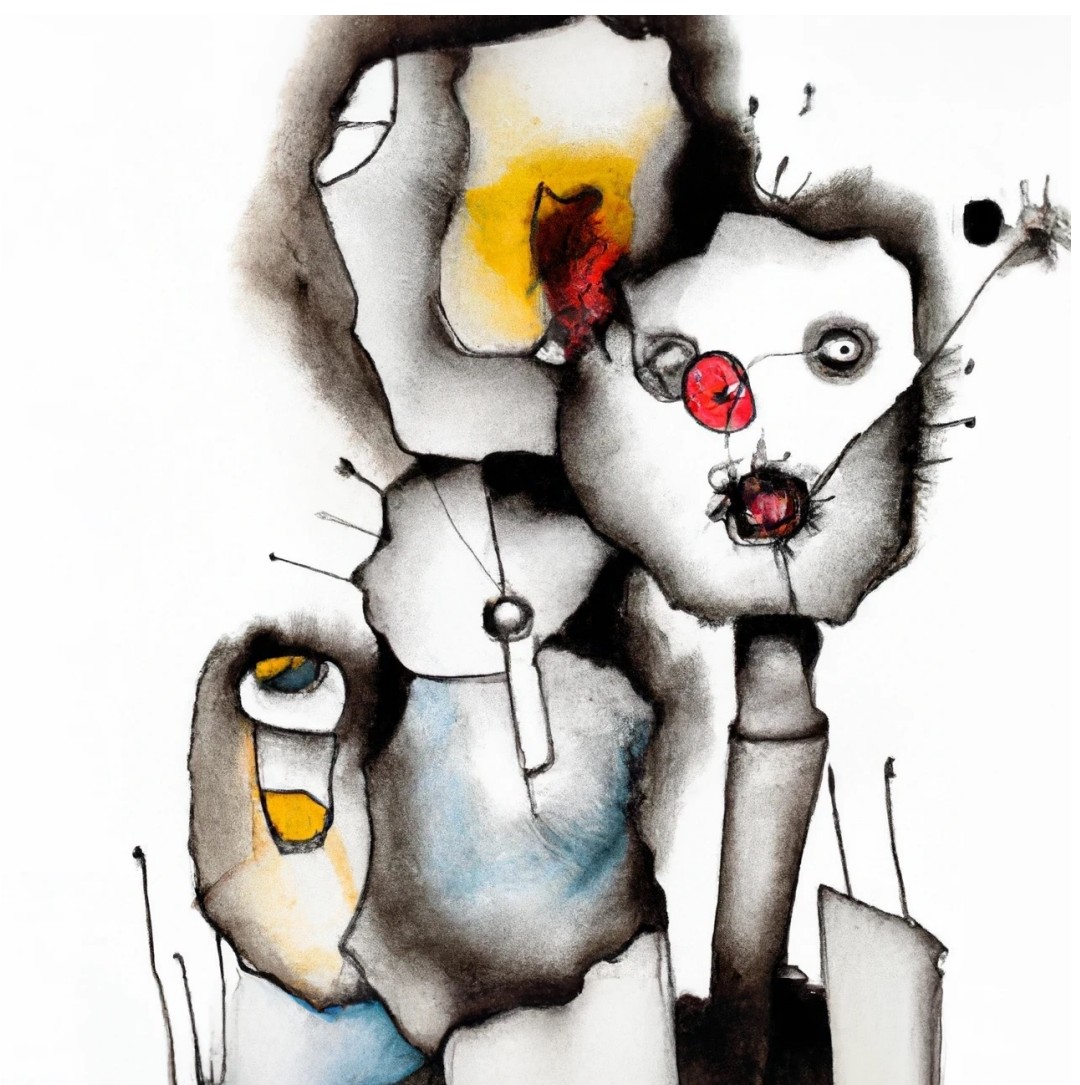

"Progress" in data visualization has been characterised by improvements in both expressiveness and effectiveness. We may be tempted to head to those distant lands of perfect expression and effect, but they will always lie beyond the horizon. Rich expressiveness must have the freedom to be ambiguous, to afford multiple interpretations, to meld with the complexities of individual and collective experience. For that is what distinguishes visualization from computation alone.

As with Gödel's incompleteness in mathematics, data visualization design is tempered by a law that cannot be subverted:

*True expressiveness can never be truly effective.*
*True effectiveness can never be truly expressive.*

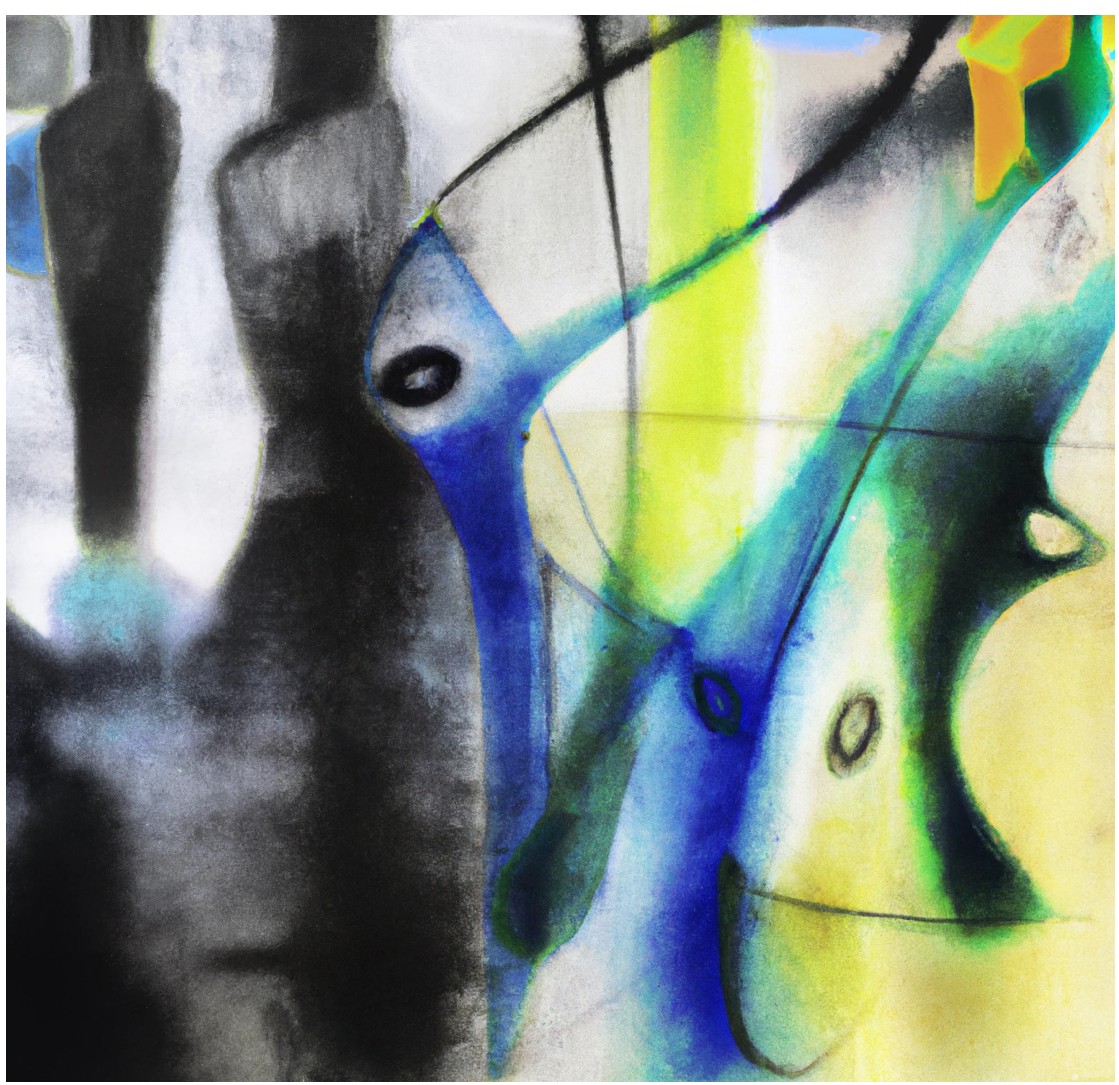

Conventional data visualization is not about to end.

But it is about to change. We can legitimately challenge the idea that AI is "intelligence", recognise that image generation encapsulates centuries of bias and inequity, and that AI text models do not meaningfully 'understand' language (Bender, 2021). Nevertheless, these technologies inspire new possibilities for the visual languages we use to convey data and for the way we construct them. Imperfect as they are, they offer us new tools with which to express ourselves.

Our walled garden remains home, but those distant lands beckon.

# Appendices

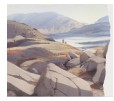 DALL·E 2 with outpainting

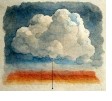 MidJourney V3

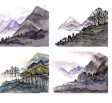 Disco Diffusion 5.2 *("angry and fearful"* and *"calm and tranquil")*

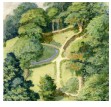 DALL·E 2

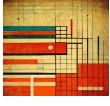 MidJourney V3

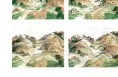 Disco Diffusion 5.2

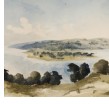 DALL·E 2 with outpainting

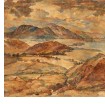 Disco Diffusion 5.6 with watercolorDiffusion 2

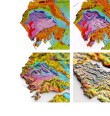 Disco Diffusion 5.2

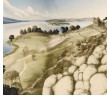 DALL·E 2 with outpainting

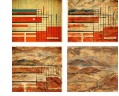 MidJourney and Disco Diffusion with watercolorDiffusion 2

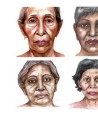 DALL·E 2

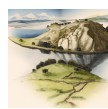 DALL·E 2 with outpainting

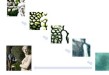 Disco Diffusion 5.2 and DALL·E 2

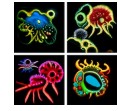 Disco Diffusion 5.3

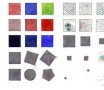 Disco Diffusion 5.6 with watercolorDiffusion 2

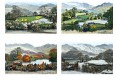 Disco Diffusion 5.2

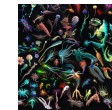 DALL·E 2 with outpainting

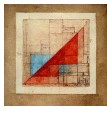 MidJourney V3

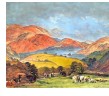 Disco Diffusion 5.2

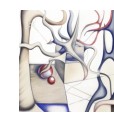 DALL·E 2 with outpainting

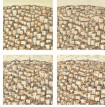 MidJourney V3 and Disco Diffusion 5.6

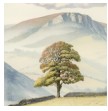 DALL·E 2 with outpainting

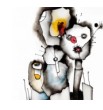 DALL·E 2

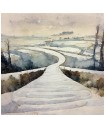 DALL·E 2

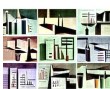 Disco Diffusion 5.2

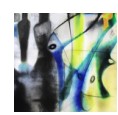 DALL·E 2 with outpainting

**Bender, E., Gebru, T., McMillan-Major, A. and Shmitchell, S.** (2021) On the dangers of stochastic parrots: Can language models be too big? 🦜. *Proceedings of the 2021 ACM Conference on Fairness, Accountability, and Transparency* pp. 610-623.

**Bertin, J.** (1983) Semiology of graphics. Esri press.

**Chernoff, H.** (1973) The use of faces to represent points in k-dimensional space graphically. *Journal of the American Statistical Association*, 68(342), pp.361-368.

**Cleveland, W and McGill, R.** (1984) Graphical perception and graphical methods for analyzing scientific data. *Journal of the American Statistical Association*, 79(387) pp.531−544.

**Kim, G., Kwon, T. and Ye, J.** (2021) DiffusionCLIP: Text-guided image manipulation using diffusion models. *arxiv.org/pdf/2110.02711.pdf*

**Mackinlay, J.** (1986) Automating the design of graphical presentations of relational information. *ACM Transactions on Graphics*, 5(2) pp.110-141
