# OpenReview forum: "Beyond the Walled Garden: A Visual Essay in Five Chapters"
_IEEE.org/2022/Workshop/altVIS — Accept_

### Official Review · Reviewer_DxRc · 2022-08-05

**Review:**

This is a very well constructed and thought provoking submission.

It's format and visual presentation is attractive, and appropriate to what it is trying to communicate.

This is the sort of work that I think should appear at Alt.VIS - it deserves an audience, and I don't know where else would be a suitable venue for it, except perhaps the VIS Arts program.

I have a few minor comments:

- p. 11, "express data by or 'sadness'" -> "express data by 'sadness'"

- p. 14, "It is the principle" -> "This is the principle"

- p. 23, there is a closing (rather than opening) quotation mark before "retinal variables"

- p. 32 it would be good to eventually state what the sentiment it is

- p. 23 "text to image technologies" -> "text-to-image technologies"

- p. 33 "data to graphic mapping" -> "mapping from data to graphic" (?)

- the references should include a DOI (or, probably better, a shortDOI: https://shortdoi.org), and ideally the DOI or arXiv ID would be a clickable link

**Conflicts:**

None

**Review Inclusion:**

Yes

**Sufficiently Alt:**

Yes

**Superlative:**

Most pastoral landscapes

---

### Official Review · Reviewer_NovQ · 2022-08-08

**Review:**

Conflict of interests:
I have no conflict of interests with the author

Signature:
Lonni Besançon

Meta Review:
-----------------

The submission has received many comments and reviews. This fact, alone, strongly suggest that it should be accepted at alt.VIS. From the submission's format to its content, the manuscript will nicely fit into the workshop and would generate many thoughts and discussions.
I would encourage the author to go through some of the suggestions (and typos) pointed in individual reviews.


Personal Review:
--------------------


The work presented in this submission is, I would argue, the best use I have seen so far of the free format offered by alt.VIS. The topic that the author tackles is very relevant to the VIS community and presented in a very visual manner and the arguments are very nicely laid out in this five-chapter visual essay.

In the second chapter, the author rightfully points out: “That expressiveness arrives bearing a cost. With a more sophisticated language, we demand more of the reader in their ability to decode the subtleties of visual languages.” Yet, I wonder if the cost of the expressiveness is not also visible in the fact that the interpretation is more likely to be subjective than in less “artistic” expressions. While I am sure we can all agree on specific characteristics that tend to make us feel one specific way about visuals/images. For instance, the work done around affective responses to image datasets (e.g., [A] or [B]) shows how subjects may interpret stimuli in very different ways.
Linked to this, I would argue that there might also be space for work that the author has not considered in the submission such as having visualizations that would respond to different emotions in order to drive them, such as done with painterly rendering techniques [C].
These two comments are merely suggestions of things I thought about while reading the work and do not necessarily have to be addressed by the author in a revision of their work.

I would thus conclude this rather-short review by stating that I am looking forward to see a presentation of the work at Alt.VIS as I am sure it will inspire the community.

Minor points:
There is a missing word in the following sentence (or the first “or” is not needed): “Instead of ‘hue’ or ‘orientation’ or ‘size’ why shouldn’t data visualization designers be able to express data by or ‘sadness’ or ‘vertigo’ or ‘naivety’?”
It would be good to say what sentiment is depicted in the image right after this quiz on page 332.


References:
[A] http://dx.doi.org/10.3758/s13428-013-0379-1
[B] https://www2.unifesp.br/dpsicobio/adap/instructions.pdf
[C] http://dx.doi.org/10.1145/1124728.1124744


**Conflicts:**

I have no conflict of interests with the author

**Review Inclusion:**

Yes

**Sufficiently Alt:**

Yes

**Superlative:**

Most visual

---

### Official Review · Reviewer_x1Xa · 2022-08-09

**Review:**

I love this. It's poetry. Please make more of this. I don't have anything else to say.

**Conflicts:**

I don't know the author

**Review Inclusion:**

No

**Sufficiently Alt:**

Yes

**Superlative:**

Most beautiful.

---

### Official Review · Reviewer_e7sH · 2022-08-09

**Review:**

This paper is a provocative idea with beautiful outputs. I'm on the fence, though, about the usefulness as a visualization approach, where the focus should be on the unambiguous interpretability of the image. For example, the sentiment on pages 29 through 32 that come to mind is "moist," which I'd wager wasn't the intention of the author. Regardless, this is a work of art and a worthy contribution to alt.VIS.

**Conflicts:**

None.

**Review Inclusion:**

Yes

**Sufficiently Alt:**

Yes

**Superlative:**

Most artistic, most moist.

---

### Official Review · Reviewer_trPB · 2022-08-24

**Review:**

Man do I have some thoughts on this one.

The fact that I have some thoughts on this one is prima facie evidence that the submission should be accepted.

But, okay, so to recap the argument:

1) Current view of vis as encoding/decoding of primitive visual variables is limiting for vis, since it
1a) focuses on discrete objects rather than holistic properties of a vis
1b) excludes potential ways of representing information (emotion, style, mood, etc.) that might permit interesting, informative, or emotionally moving visualizations.
2) Current text2image systems like DALL-E/MidJourney allow the “steering” of images based on things like mood or style.
3) And therefore we could use these ML techniques to expand the space of data visualization.

I generally agree with 1) + 1a). I even wrote a whole paper with Enrico and Steve about how much I agree with 1): https://arxiv.org/pdf/2008.11310.pdf

I also agree with 1b). I think there are great opportunities to incorporate style, affect, and other properties not just as artistic constraints but also rhetorical or even data-representing components of a visualization. Your very own “Sketchy Rendering for Information Visualization” comes to mind here: what if the uncertainty or dubious provenance of a data visualization was directly represented by the holistic stylistic choices of the data? A shaky pencil sketch on a cocktail napkin that gradually becomes more and more refined and “professional” as data comes in and uncertainties are reduced. An artist outside of the “walled garden” could probably blow us away here in our Bertin-Tufte axis of minimalism encoding of data into glyphs. Hell, Giorgia Lupi does stuff with affect and style even within glyphsville.

It’s 3) where I start to challenge things. I suspect I am being a Luddite about this, but for me I’m always a bit uneasy with all of these attempts to show how cool text2image ML is, or how it could automate important design problems for us (beyond the design problem of “how to make occasionally cool images with particular properties).

My concerns are:
1) It’s “AI washing” that is reliant on the hard work of a large set of artists but offers them no recognition or payment.
2) It’s a “corpus reguritator” that can instantiate the biases of its source corpus. This is especially a problem if, as in this application, you’re hoping to associate data with some adjective, and that adjective is hopelessly entangled with some pretty poor behavior. Would you like if you encoded data as the “goodness” or “evilness” of an image, and that resulting notion of “good” and “evil” ended up just portraying a racist, sexist, or fetishistic vision of the world?  (see https://reticular.hypotheses.org/5216 which undermines my point 1) above but has a very nuanced, if occasionally crass, discussion of this point)
3) The “encoding” metaphor is sometimes useful, since at least we know what we are getting, and how to use it. Or, maybe more broadly, having some legible+deterministic process for generating a visualization based on input data is useful. If I have to use these visualizations for actual use, the fact that slight differences in prompts, let alone the drastic differences in training corpora or algorithm in use, can produce dramatically different visualizations, is a problem. Maybe even an algebraic visualization design-scale “hallucinator” problem: I see very different visualizations, so I assume that my data are very different, but it’s just because I used a different seed than somebody, or capitalized a word in my text prompt. I haven’t learned about my data, I’ve just learned (a very narrow) slice about how the text2image algorithm parses prompts.

In short, if you had made almost the exact same submission, but just told me you hand painted all of these, or that you grabbed them off of google search, I would be far less uneasy with what is being proposed than what I got. That perhaps says more about my psychology than the submission, but I thought it was worth mentioning.

**Conflicts:**

I don't believe I'm conflicted formally.

**Review Inclusion:**

Yes

**Sufficiently Alt:**

Yes

**Superlative:**

Most Meditative

---

### Official Review · Reviewer_w2WP · 2022-08-25

**Review:**

The author provides a beautiful brochure for a collection of AI-assisted visualizations. The authors call for a more holistic view of the metrics for visualizations, especially in considering the sentiments of the illustrations beyond the conventional metrics.

However, the paper does not fit the conference submission criteria? I believe the works are ground-breaking in integrating psychology and statistics, art and design but not sure whether it fits the part of the conference. Maybe better to appear in a high-scale meta-verse art-work exhibition?

**Conflicts:**

NA

**Review Inclusion:**

No

**Sufficiently Alt:**

Yes

**Superlative:**

Most artistic

---

### Decision · Program_Chairs · 2022-08-31

Accept